



# The internal structure and composition of a plate boundary-scale serpentinite shear zone: The Livingstone Fault, New Zealand

Matthew S. Tarling[1], Steven A.F. Smith[1], James M. Scott[1], Jeremy S. Rooney[2], Cecilia Viti[3], and Keith C. Gordon[2]

[1]Department of Geology, University of Otago, 360 Leith Street, 9016 Dunedin, New Zealand
[2]Department of Chemistry, University of Otago, Union Place West, 9016 Dunedin, New Zealand
[3]Dipartimento di Scienze Fisiche, della Terra e dell'Ambiente, Università degli Studi di Siena, Siena, Italy

**Correspondence:** Matthew S. Tarling (tarlingmatthew@gmail.com)

**Abstract.** Deciphering the internal structure and composition of large serpentinite-dominated shear zones will lead to an improved understanding of the rheology of the lithosphere in a range of tectonic settings. The Livingstone Fault in New Zealand is a >1000 km long terrane-bounding structure that separates the basal portions (peridotite; serpentinised peridotite; metagabbros) of the Dun Mountain Ophiolite Belt from quartzofeldspathic schists of the Caples or Aspiring Terranes. Field and microstruc-
tural observations from eleven localities along a strike length of c. 140 km show that the Livingstone Fault is a steeply-dipping, serpentinite-dominated shear zone tens to several hundreds of metres wide. The bulk shear zone has a pervasive scaly fabric that wraps around fractured and faulted pods of massive serpentinite, rodingite and partially metasomatised quartzofeldspathic schist up to a few tens of metres long. S-C fabrics and lineations in the shear zone consistently indicate a steep Caples-side-up (i.e. east-side-up) shear sense, with significant local dispersion in kinematics where the shear zone fabrics wrap around pods.
The scaly fabric is dominated (>98 vol%) by fine-grained (<<10 $\mu$m) fibrous chrysotile and lizardite/polygonal serpentine, but infrequent (<2 vol%) lenticular relics of antigorite are also preserved. Dissolution seams and foliation surfaces enriched in magnetite, as well as the widespread growth of fibrous chrysotile in veins and around porphyroclasts, suggest that bulk shear zone deformation involved pressure-solution. Syn-kinematic metasomatic reactions occurred along all boundaries between serpentinite, schist, and rodingite, forming multi-generational networks of nephritic tremolite veins that are interpreted to have
caused reaction-hardening within metasomatised portions of the shear zone. A general conceptual model is proposed for the internal structure and composition of plate boundary-scale serpentinite shear zones deforming at greenschist-facies conditions. The model involves bulk distributed deformation by pressure-solution creep, accompanied by a range of physical (e.g. faulting in pods and wall rocks; smearing of magnetite along fault surfaces) or chemical (e.g. metasomatism) processes that result in localised brittle deformation within creeping shear zone segments.

## 1 Introduction

The mechanical and seismological characteristics of serpentinite-bearing shear zones are controlled by shear zone structure
and composition. Because serpentinite influences the rheology of the lithosphere in a range of tectonic settings, substantial effort has been aimed at collecting experimental (e.g., Brantut et al., 2016; Reinen et al., 1991, 1994; Moore et al., 1996; Kohli



et al., 2011; Auzende et al., 2015; Tesei et al., 2018), geochemical (e.g., Viti and Mellini, 1997; Bebout and Barton, 2002) and petrological (e.g., Coleman, 1971; Viti and Mellini, 1998; Auzende et al., 2002) data from serpentine minerals and serpentinite fault rocks. Much of this work has been summarised in a series of recent review papers (Guillot and Hattori, 2013; Hirth and Guillot, 2013; Reynard, 2013; Guillot et al., 2015; Viti et al., 2018). However, extrapolation and up-scaling of such data to

natural serpentinite shear zones remains problematic because: 1) serpentinite shear zones are structurally and lithologically complex (Strating and Vissers, 1994; Hermann et al., 2000; Singleton and Cloos, 2012), and their physical properties are therefore not adequately represented by laboratory measurements that are often made on small monomineralic samples, and; 2) serpentine itself has several sub-types that are challenging to identify unambiguously, particularly in a way that preserves the typically intricate textural relationships between the sub-types. A better understanding of how to apply laboratory data to

natural serpentinite shear zones requires improvements to be made on several fronts. This should include effort to document in high-resolution the internal structure and composition of well-exposed natural serpentine shear zones across a wide range of scales and metamorphic facies, which will allow interpretations to be made regarding the temporal and spatial evolution of shear zone rheology.

Currently, there are only a few detailed field descriptions of the internal structure and composition of large serpentinite

shear zones (e.g., Norrell et al., 1989; Strating and Vissers, 1994; Hermann et al., 2000; Soda and Takagi, 2010; Singleton and Cloos, 2012). This limits our understanding of the potential role that serpentinite-bearing shear zones may play in controlling processes such as deep episodic tremor and slip (ETS; Poulet et al., 2014), decoupling and weakening of the subducting slab and mantle wedge (e.g., Hirauchi et al., 2013; Moore et al., 2004; Moore and Lockner, 2013), and exhumation of high-pressure metamorphic rocks in subduction channels (Federico et al., 2007; Hermann et al., 2000). Field-based studies of large

serpentinite shear zones typically document a pervasive anastomosing foliation in the bulk of the shear zone (Norrell et al., 1989; Strating and Vissers, 1994; Hermann et al., 2000; Soda and Takagi, 2010; Singleton and Cloos, 2012), commonly referred to as "scaly" fabric - a descriptive term used to characterise rocks displaying a phacoidal cleavage at the micro and macroscale (e.g., Moore, 1986; Vannucchi et al., 2003; ?). Some studies have highlighted the potential importance of pressure-solution in the development of this fabric at greenschist-facies conditions (Andréani et al., 2005, 2004). Other studies have focused on the

role of metasomatism in areas where serpentinite is in contact with chemically-distinct rock types (Moore and Rymer, 2007; Soda and Takagi, 2010). However, field studies are often hampered by relatively poor exposure in areas dominated by scaly serpentinite fault rocks, and it is rare to have the internal structure and composition of a plate boundary-scale serpentinite shear zone exposed in detail over expansive areas.

The purpose of this paper is to present field and microstructural observations of the internal structure and composition of a

plate boundary-scale serpentinite shear zone that is well exposed in the South Island of New Zealand. The Livingstone Fault represents an important opportunity to document the characteristics of a large serpentinite-dominated shear zone from the sub-micron scale up to the crustal scale. We present structural and petrological data on the geometry, kinematics and composition of the shear zone. These data are used to propose a general conceptual model that could be used as a framework to help interpret the mechanical behaviour, and seismological and physical properties, of active serpentinite-bearing shear zones.





## 2 Geological Setting and Previous Work

The continent of Zealandia is composed of a series of Cambrian to Cretaceous tectonostratigraphic terranes that were amalgamated on the paleo-Pacific Gondwana margin (Fig. 1). The Zealandia terranes comprise an Early Cretaceous fore-arc, arc, and a portion of the back-arc, which were rifted from Australia and Antarctica after c. 84 Ma (e.g., Gaina et al., 1998). The

terranes are broadly subdivided into the Western Province and Eastern Province, which are separated by the Median Batholith (e.g., Mortimer, 2004; Landis and Coombs, 1967; Bishop et al., 1985, ; Fig. 1). The Western Province is dominated by Early Palaeozoic metasedimentary rocks (e.g., Cooper, 1989; Adams et al., 2015) and was the focus of arc magmatism for 400 Ma (e.g., Kimbrough et al., 1994; Muir et al., 1996; Tulloch et al., 2009; Allibone et al., 2009). The Eastern Province is dominated by Late Palaeozoic to Mesozoic metasedimentary terranes (e.g., MacKinnon, 1983; Adams et al., 1998) that were progressively

accreted above the subduction margin. Sandwiched within the Eastern Province metasedimentary terranes is a Permian-Triassic island arc (Brook Street Terrane; Landis et al., 1999, ; Fig. 1) and an ophiolite belt (Dun Mountain-Maitai Terrane, referred to commonly as the Dun Mountain Ophiolite Belt, DMOB; Fig. 1; Coombs et al., 1976, ; Fig. 1). The Cretaceous boundary between the Western Province and Eastern Province is a complex zone of arc magmatism that intrudes both Provinces and is referred to as the Median Batholith (Mortimer et al., 1999). A series of major ductile shear zones mark a precise Western

Province-Eastern Province boundary in some places (Bradshaw, 1993; Scott, 2013; Allibone and Tulloch, 2008). Most terrane boundaries, however, appear to have been reactivated by Cenozoic faulting associated with development of the present-day Australia-Pacific plate boundary system.

The Livingstone Fault is the terrane boundary that defines the eastern margin of the DMOB (Figs. 1, 2). The DMOB is a narrow slice (<20 km) of Permian oceanic lithosphere that can be traced for more than 1000 kilometres through the North and

South Islands of New Zealand, although it has been offset by 480 km across the Alpine Fault in the last c. 25 Ma (Wellman, 1953; Sutherland et al., 2000). The DMOB is well exposed in the South Island of New Zealand. In the North Island it is buried underneath relatively young volcanic and sedimentary rocks, except in one location at Piopio (Fig. 1). However, it can be traced through the North Island using the strong positive magnetic anomaly that is formed by magnetite-bearing serpentinites (Coombs et al., 1976; Eccles et al., 2005). A complete ophiolite sequence that includes an ultramafic basal section (harzburgite to dunite,

and serpentinised equivalents), a mafic dyke complex, a mafic volcanic sequence, and an overlying sedimentary sequence (the Maitai sequence, including: limestones, conglomerates, sandstones, mudstones, volcaniclastics) is preserved at Red Mountain in South Westland (Fig. 2) and at Dun Mountain near Nelson (Coombs et al., 1976; Sinton, 1977; Sano et al., 1997; Stewart et al., 2016). During accretion, the ophiolite sequence was rotated so that the bulk layering is now subvertical and the overall sequence youngs from east to west (Fig. 2; Eccles et al., 2005; Gray et al., 2007). Uranium-Pb zircon dates from plagiogranites

in the upper portion of the layered gabbros and dike complex suggest a formation age of 275 - 285 Ma (Kimbrough et al., 1992), which is consistent with detrital zircons in the sedimentary Maitai component (Jugum et al., 2013)). The DMOB is situated between zeolite facies volcanogenic sediments of the Murihiku Terrane to the west, and prehnite-pumpellyite to lower greenschist facies schists of the Caples Terrane and Aspiring Terranes to the east (Figs. 1, 2).



The DMOB and Livingstone Fault are well exposed in intermittent locations through a ~150 km-long strip in the South Westland area of the Southern Alps, which is the focus of this paper (Figs. 1, 2). Early work mentioning the Livingstone Fault consisted mainly of regional mapping projects focused on the adjacent Murihiku and Caples/Aspiring Terranes (Bishop et al., 1976; Cawood, 1987, 1986; Coombs et al., 1976; Craw, 1979; Grindley, 1958; Hutton, 1936; Macpherson, 1946; Turnbull, 1980; Wood, 1956). Grindley (1958) and Bishop et al. (1976) noted that the fault has a steeply east-dipping or sub-vertical orientation. The most comprehensive work to date was by Craw (1979), who recognised the fault to be a sub-vertical to steeply east-dipping serpentinite-bearing shear zone up to 200 m wide near West Burn (Fig. 2). Later work by Cawood (1987, 1986) in the central Southland area recognised a steeply-dipping fault characterised in places by a schistose serpentinite matrix containing blocks of quartzofeldspathic lithologies, massive serpentinite and rodingite (meta-gabbro/-dolerite) up to 5 metres long. Interpretations of the South East South Island (SESI) geophysical transect suggest that the Livingstone Fault probably extends at least to the base of the crust at 20-30 km depth (Fig. 1b) (Mortimer et al., 2002). Cawood (1986) interpreted that the most recent phase of deformation involved a steep reverse sense of motion, with the Caples and Aspiring Terranes having been thrust over the DMOB. However, the timing of deformation events is poorly constrained because the shear zone assemblages are not conducive to dating. It seems likely that the Livingstone Fault has experienced multiple phases of reactivation, including possible Cenozoic reactivation where it lies subparallel to the Alpine Fault. The fault must have been active post-Jurassic because these are the youngest sedimentary strata found in the adjacent terranes (Cawood, 1986).

## 3 Methods

### 3.1 Fieldwork and drone-assisted mapping

Detailed field mapping, sampling and structural data collection were performed at 12 locations along a strike length of c. 140 km (Fig. 2). Around 5 km to the NE of Cosy Gully, the Livingstone Fault is truncated by the Alpine Fault. In several locations (stereonets in Figure 2), the asymmetry of well-defined S-C fabrics, combined with lineation measurements, allowed the bulk shear sense to be determined. Orientated samples were collected that represent the main shear zone, wall rock, and metasomatic lithologies, and the most important macroscale structural characteristics of the shear zone were noted.

Drone imagery and photogrammetry were used to produce a high-resolution orthorectified aerial photo of the Livingstone Fault at Serpentine Saddle (Fig. 2), which was used as a basemap to survey the internal structure of the shear zone at this locality. Aerial photography was performed using a DJI Phantom 4 Pro drone (DJI, Shenzhen, China), which was manually piloted at an average height of approximately 55 metres above ground level. GPS-tagged pictures were taken at a semi-regular interval in order to obtain approximately 30% lateral and 60% forward overlap between adjacent photos. A total of 615 pictures were taken, covering a total area of approximately 0.3 km$^2$. The photographs were processed using Agisoft PhotoScan software to create a single orthorectified aerial photo with a ground resolution of c. 2.15 cm per pixel (See Supplementary Information Figure 1 for the high-resolution orthophoto). Image analysis with ImageJ software (Schneider et al., 2012) was used to quantify the abundance of shear zone lithologies after mapping had been completed.



## 3.2 Microscopy and microstructural observations

Thin sections of fault rocks were cut parallel to lineation and perpendicular to foliation. Standard 30-$\mu$m thick polished petrographic thin sections were prepared for microstructural observations. Observations were carried out using a combination of transmitted and reflected-light optical microscopy, Raman spectroscopy mapping (section 3.3), scanning electron microscopy

(SEM) and transmission electron microscopy (TEM). SEM imaging was performed using a Zeiss Sigma VP Field-Emission Scanning Electron Microscope at the Otago Centre for Electron Microscopy. The SEM was operated using an accelerating voltage of 15 kV and a working distance ranging from 6 to 8.5 mm. TEM samples were extracted from thin sections prepared using Canada balsam adhesive, by mounting copper annuli grids 3 mm in diameter with a central hole of 800 $\mu$m diameter. An Ar+ ion milling precision polishing system (PIPS) was used to mill the samples to electron transparency (Gatan Inc., United

States). TEM analysis was performed on a JEOL JEM-2010 microscope (JEOL Ltd., Tokyo, Japan) at the University of Siena, Italy. The TEM was operated at 200 kV with a LaB6 source and ultra-high resolution pole pieces, resulting in a point resolution of 0.19 nm.

## 3.3 Raman spectroscopy

Thin sections prepared for Raman spectroscopy were glued using a low-fluorescence epoxy (Epofix Cold-Setting Embedding

Resin, Struers) (Rooney et al., 2018). Raman spectroscopy mapping was performed on an Alpha 300R+ confocal Raman microscope (WITec GmbH, Ulm, Germany) in the Chemistry Department at the University of Otago, New Zealand. A dry 100x objective (Carl Zeiss AG, Oberkochen, Germany), 1200 g mm$^{-1}$ grating and a 532 nm wavelength laser (Coherent, Santa Clara, California) at c. 50 mW were used. The laser spot size and spatial resolution of this setup is approximately 370 nm (Rooney et al., 2018). A piezo-controlled nanopositioning stage was used to control the sample position during the Raman

mapping process. The Raman microscopy system was calibrated with a semiconductor-grade silicon wafer using the 520.6 cm$^{-1}$ band, followed by verification of the 3620.6 cm$^{-1}$ band from a sample of kaolinite. WITec Project Plus software was used to analyse the Raman data to produce colour maps based on the spatial distribution of the Raman signal of the minerals present. Full details regarding the collection and processing of sub-micron Raman spectroscopy data are available in Rooney et al. (2018) and Tarling et al. (2018a).

## 25  4  Results

### 4.1  Regional structure and kinematics of the Livingstone Fault

The Livingstone Fault consists of a serpentinite-dominated shear zone that separates the mainly quartzofeldspathic schists of the Caples or Aspiring Terranes from mafic or ultramafic portions of the DMOB (Fig. 3). In many of the examined localities (i.e. Fiery Col, Cow Saddle, Mount Raddle, Mount Richards. Red Spur), the DMOB wall rocks are peridotite and serpentinised

peridotite (Fig. 2). However, at Serpentine Saddle (section 4.2), Four Brothers Pass and Beresford Pass, ultramafic portions of the ophiolite are absent and the serpentinite shear zone separates schists from metagabbros and associated dykes. The thickness



of the shear zone varies significantly along strike (Fig. 2). For example, at Fiery Col (Fig. 3a,b) and Cow Saddle (Fig. 3c), the shear zone is 50-90 m wide (similar to West Burn, East Eglinton River, Mount Raddle and Red Spur; Fig. 2). At Serpentine Saddle the shear zone is up to 420 m wide, whereas at Mount Richards (25 m) and Cosy Gully (5 m) it is much narrower.

Boundaries between the serpentinite shear zone and the wall rocks are commonly steeply dipping and well defined (Fig. 3). In locations where the DMOB wall rocks are peridotite or serpentinised peridotite, the western boundary of the shear zone is defined by a progressive transition, over a distance of a few metres to tens of metres, from scaly shear zone serpentinite to partially serpentinised peridotite and then into peridotite (Fig. 3c). At Serpentine Saddle, where the DMOB wall rocks are metagabbros, late-stage faults offset and disrupt the western shear zone boundary. The eastern shear zone boundary is typically defined by a network of steeply-dipping brittle faults within and at the contact with the schist host rocks (Fig. 3c, 4). These faults surround lenses of partially metasomatised and fractured schist up to hundreds of metres long and tens of metres wide (Fig. 4a). The faults have polished, slickensided surfaces (Fig. 4b) associated with well-cemented cataclasite layers up to 2 cm thick. The faults are mainly sub-vertical to steeply west-dipping and lineations plunge sub-vertically (Fig. 4b).

All exposures of the serpentinite shear zone are characterised by a strongly foliated matrix with a scaly fabric, defined by sub-cm asymmetric phacoids of serpentinite (Fig. 5). This fabric wraps around pods of rodingite, quartzofeldspathic schist, massive serpentinite and veined serpentinite ranging from <cm to tens of m long (Figs. 3b, 5, 7). The surfaces of serpentinite phacoids are often polished and contain lineations defined by surface striae (i.e. grooves) or bundles of aligned serpentine mineral fibres. Overall, the scaly foliation is steeply-dipping and subparallel to the regional orientation of the Livingstone Fault (Fig. 2). In several localities (green and blue data on stereonets on Figure 2), the scaly foliation is represented by a well-defined "S-C" fabric (Figs. 2, 5; Berthé et al., 1979; Lister and Snoke, 1984; Passchier and Trouw, 2005). The intersections between the "S" foliation surfaces and the "C" shear bands plunge shallowly in a direction subparallel to the strike of the shear zone boundaries (Fig. 2). The "C" shear bands are relatively planar, continuous for up to tens of centimetres, and deflect the "S" fabric (Fig. 5). There is an angle of 25 – 40° between the S and C surfaces (Fig. 2). Both the internal and external asymmetry of the S-C fabric (as defined in Passchier and Trouw (2005); Fig. 5), combined with the dominance of moderately-to-steeply plunging shear zone lineations (Fig. 2), consistently indicate an overall Caples/Aspiring Terrane-up shear sense (i.e. east side-up).

At Cow Saddle, pods of massive serpentinite are cut by en echelon brittle faults that have a similar orientation to the "C" shear bands in the surrounding matrix (Figs.2, 5). These faults contain steeply-plunging lineations (Fig. 2) and show offsets that are compatible with the overall east-side-up kinematics inferred from the S-C fabrics (Figs. 2, 5). Additionally, at Cow Saddle and Fiery Col, much larger pods of schist up to 200 m long and up to 60 metres wide are completely surrounded by serpentinite, indicating that they were probably fault-bound lenses of Caples/Aspiring-Terrane (such as those shown in Figure 4a) that were plucked off and incorporated in to the serpentinite shear zone.

The Livingstone Fault contains metasomatic reaction zones in a number of structural locations, and the reactions zones are ubiquitous at all the investigated localities shown on Figure 2. These reactions zones are described in more detail in Tarling et al. (In Revision), and only a brief description of the most important characteristics is provided here. Along the main shear zone boundary that separates serpentinite from the Caples Terrane schists, metasomatic reaction zones are characterised by multigenerational tremolite (and minor talc and diposide) vein networks that form within tabular layers up to several tens of



metres wide (Fig. 6). Metasomatic hydrogrossular, andradite garnet and clinochlore are found as accessory minerals both within the veins and in the adjacent serpentinite. Additionally, pods of schist, rodingite and partially-rodingitised gabbro/dolerite within the shear zone can be surrounded by complex tremolite vein networks up to several metres wide (Fig. 6a). Overall, the metasomatic veins lack any strong preferred orientation, although in some places the earliest vein set appears to have exploited

the serpentinite foliation. Where offset of pre-existing veins can be seen, there is no evidence for a significant shear component, and most of the veins appear to have been tensile in nature. In areas of the shear zone that are severely metasomatised, the vein networks and surrounding serpentinite are cut by discrete, cataclastic fault surfaces that are steeply dipping and contain steeply-plunging lineations (Fig. 6b), consistent with the kinematics determined from the scaly foliation and from fault surfaces that cut through pods.

## 4.2 Internal structure and composition of the Livingstone Fault at Serpentine Saddle

### 4.2.1 Overall shear zone structure and wall rocks

The shear zone at Serpentine Saddle is up to 420 metres thick (Figs. 7, 8a). The DMOB wall rocks here consist mainly of metagabbros and associated dykes; no wall rock peridotite or serpentinised peridotite is present in this locality. The Caples Terrane wall rocks on the eastern side are dominated by greenschist-facies metamorphic assemblages containing quartz, pla-

gioclase, chlorite, epidote and muscovite (Bishop et al., 1976; Turnbull, 1980). The western boundary of the shear zone is irregular due to the presence of late-stage brittle fault surfaces that offset the boundary between the shear zone and the basal DMOB wall rocks (Fig. 7). The eastern shear zone boundary is defined by a region up to 20 m wide containing extensively metasomatised Caples Terrane schists, as well as networks of metasomatic veins.

The shear zone in the mapped area consists of (Fig. 7):

1. ∼ 80% scaly serpentinite that in places has a well-defined S-C fabric (stereonet i);

2. ∼ 18% massive serpentinite, which occurs either as fractured and boudinaged pods up to c. 20 m long (stereonet ii), or in much larger domains that have gradational boundaries with the surrounding scaly serpentinite;

3. ∼ 2% rodingite, which occurs either as fractured and boudinaged pods up to 30 m long, or as relatively planar, shallowly-dipping dykes/sills within massive serpentinite domains (stereonet iii);

4. <1% schist, which is mainly represented by a single pod close to the eastern shear zone boundary, and;

5. <0.01% chromitite pods up to a few metres in size that are dispersed throughout the shear zone and are typically bound by highly-polished cataclastic fault surfaces.

### 4.2.2 Massive serpentinite domains and pods in the shear zone

Large domains of massive serpentinite tens to hundreds of metres long (Fig. 7) are characterised by a negligible to weak

fabric, as well as preservation of mesh-textured serpentinite (section 4.3). Sub-horizontal to shallowly-dipping rodingite dykes





are preserved in these domains (Figs. 7, 8b). Outside the massive serpentinite domains, the rodingite dykes are progressively rotated into alignment with the steeply-dipping scaly foliation, resulting in fracturing and boudinage of the dykes to form isolated pods surrounded by serpentinite (Figs 7, 8c), or chains of aligned elongate pods (Figs 7, 8d,e). The margins of some rodingite pods are sheathed in a metasomatic 'blackwall' rim, consisting mainly of monomineralic, fine-grained chlorite (Fig.

8f). In places, the blackwall rim is sheared off or disrupted by the scaly fabric, putting the rodingite in direct contact with the scaly serpentinite. In these cases, networks of nephritic tremolite veins are found along the contacts between rodingite and serpentinite. The outer margins of the rodingite pods are commonly defined by polished slickensides containing shallowly- to steeply-plunging lineations (stereonet iii, Fig. 7), with underlying cataclastic layers up to tens of cm thick. In detail, the margins of rodingite (and massive serpentinite) pods show two preferred orientations: 1) NW-SE striking and steeply NE or

SW dipping, and 2) E-W striking and moderately- to steeply-N or S-dipping (stereonet iii, Fig. 7). These preferred orientations highlight the characteristic shape of many of the pods: in map view, pods often have a lenticular to rhomboidal shape, with the long-axis of the pod (in map view) subparallel to the strike of the surrounding scaly foliation and the margins of the shear zone. Although outcrop constraints limited the investigation of shear zone geometry in cross-sectional view, pods that could be observed in three dimensions also showed a slightly asymmetric lenticular or rhomboidal shape in cross-section (Fig. 8f).

The asymmetry of the pods in cross-section is compatible with the Caples-Terrane-up (east-side-up) shear sense indicated by the S-C fabrics in the scaly serpentinite.

Pods of serpentinite contain a central core of massive serpentinite surrounded by an outer cladding that transitions towards scaly serpentinite (Fig. 8g). The massive serpentinite pods are typically heavily fractured with a range of fault and lineation orientations (stereonet ii, Fig. 7). In one area of the map, two large pods of massive serpentinite are observed to be in direct

contact with one another, interpreted to result from "collision" between the two pods during shearing (Figs. 7, 8d, e). In this location, the contact zone between the two pods contains a dense network of fractures that radiate outwards from the area of contact into the centre of the pods (Fig. 8d,e).

Another type of resistant pod observed in the shear zone consists of moderately foliated serpentinite containing embedded fragments of partially rodingitised gabbroic to doleritic dykes (cm to 10s cm wide by 10s cm to m length). Network of tremolite

veins radiate out from the dyke fragments and crosscut the surrounding serpentinite foliation.

### 4.2.3 Scaly foliation and S-C fabrics

The S-C fabrics and associated lineations at Serpentine Saddle indicate an overall Caples Terrane-up shear sense (east-side up), consistent with other localities (Figs. 2, 7). However, in detail there is substantial spread in the orientations of the scaly foliation, S-C fabrics, and lineations, due to deflection around pods and massive serpentinite domains (stereonet i in Fig. 7)

(Vannucchi et al., 2003; **?**). The mean orientation of the S fabric is 172°/68°W, but with 80° of strike dispersion and 40° of dip dispersion exhibited by the main cluster of data points (stereonet i in Fig. 7). The mean orientation of the C shear bands is 014°/79°E, with 90° of strike dispersion and 35° of dip dispersion (stereonet i in Fig. 7).





### 4.3 Composition and textural evolution of the serpentinite shear zone

#### 4.3.1 Massive serpentinite domains and pods

Massive serpentinite within pods contains pseudomorphic bastite and mesh textures, suggesting that these regions represent relatively undeformed serpentinised peridotite (Viti and Mellini, 1998) (Fig. 9a). Bastites are textural pseudomorphs of py-
roxene minerals, while mesh textured serpentine forms as the result of the serpentinisation of olivine grains (Viti and Mellini, 1998; Wicks and Whittaker, 1977). Serpentinisation of the original olivine and pyroxene typically produces abundant magnetite that is initially disseminated throughout the pseudomorphic serpentinite as sub-micron grains (Figs. 9, 10; O'Hanley and Dyar, 1993). Around the edges of massive serpentinite pods and domains, pseudomorphic serpentinite shows textural evidence for the onset of shearing (Fig. 9b). The mesh textures show signs of developing into 'ribbon textured' serpentinite, where
portions of the mesh structure are preferentially dissolved to form elongate lensoid ribbons (Fig. 9b) (Viti et al., 2018). In such areas, porphyroclastic magnetite (Fig. 9b) and relict chromite grains are also common. Partial dissolution of the mesh textured serpentinite results in some initial concentration of magnetite around the boundaries of the deforming meshes. Magnetite-rich seams are continuous along these boundaries for at least several millimetres (Fig. 10a).

#### 4.3.2 Scaly shear zone serpentinite

The most common scaly serpentinite in the shear zone consists of phacoids of serpentinite that have rounded edges and sigmoidal shapes that contribute to the asymmetry of the S-C fabric (Fig. 5). The slip surfaces that make up the exterior of these lenses are commonly coated in a shiny, polished serpentinite, although in some outcrops weathered coatings of fibrous chrysotile give the serpentinite a rougher, splintery appearance. The scaly serpentinite displays a 'fractal-like' geometry in which each phacoid can be cleaved apart to create smaller phacoids of serpentinite, a common characteristic of scaly fabric
fault rocks (Maltman et al., 1997; Vannucchi et al., 2003; **?**). This self-similarity in geometry is observed down to a scale of <10 $\mu$m, where lenticular domains of serpentinite are separated by shear planes along which seams of magnetite are concentrated (Fig. 10).

The shape of individual phacoids and the spacing of the foliation surfaces can vary substantially throughout the shear zone. In samples of scaly serpentinite that have relatively widely-spaced foliation surfaces, deformed mesh and ribbon-textured ser-
pentinite are preserved inside phacoids, and the phacoids are coated by continuous and interconnected seams of magnetite (Fig. 10a). Where the foliation surfaces are more closely spaced, phacoids of serpentinite contain no evidence for the preservation of pseudomorphic textures (Fig. 10b). Instead, lenticular domains of fine-grained chrysotile, lizardite and polygonal serpentine are outlined by interconnected seems of magnetite and fibrous chrysotile (Fig. 10b). In scaly serpentinite with lower proportions of magnetite, discontinuous seams of magnetite are concentrated along the boundaries of the scaly foliation, broadly defining
the overall lenticular shape of serpentinite phacoids (Fig. 10c).

A combination of TEM and Raman spectroscopy mapping reveals that the scaly serpentinite is composed of fibrous chrysotile (70-80 wt.%; Figs. 11a, b, c), lizardite (10-25 wt.% Figs. 11b, d), minor polygonal serpentine (Fig. 11c) and magnetite (Tarling et al., 2018a; Rooney et al., 2018). Relict chromite grains are dispersed throughout the serpentinite and are typically mantled





by a layer of ferritchromit and occasionally surrounded by chloritic aureoles, which are thought to form by a dissolution-precipitation mechanism (Mellini et al., 2005). Minor brucite occurs in association with sheared lizardite and chrysotile. Secondary phases dispersed as small grains throughout the scaly serpentinite include awaruite, wairauite, pentlandite, millerite, heazlewoodite, native copper and copper oxides.

Aggregates of antigorite are present within small (10-400 $\mu$m), isolated porphyroclasts distributed throughout the scaly serpentinite, but make up <1 vol. % (Figs. 11a,b). The porphyroclasts consist of interpenetrating blades of antigorite, a texture that has previously been interpreted as characteristic of prograde metamorphic serpentinites (Wicks and Whittaker, 1977; Wicks, 1984; Viti et al., 2018). The porphyroclasts are surrounded by a matrix of chrysotile, fine-grained lizardite and/or polygonal serpentine (these two varieties are indistinguishable with Raman; Fig. 11b; Tarling et al., 2018). The long axes of

the porphyroclasts are subparallel to the scaly foliation, and the porphyroclasts are "truncated" along their foliation-parallel margins by surfaces that are enriched in thin layers or aggregates of lizardite/polygonal serpentine (Fig. 11b). Chrysotile preferentially grows in fine-grained, fibrous 'beards' around the ends of the porphyroclasts (Figs. 11a, b).

## 5   Discussion

### 5.1   Deformation conditions within the Livingstone Fault

The dominance of crystalline lizardite and fibrous chrysotile, together with the apparent instability of antigorite, is consistent with an estimated ambient temperature during shearing of 300-350 °C. This is also broadly compatible with interpretations of prehnite-pumpellyite to lower greenschist facies metamorphic facies in the Caples/Aspiring Terrane wall rocks (Bishop et al., 1976; Turnbull, 1980). While pressure estimates are difficult to obtain due to the inherently pressure-insensitive nature of the serpentine minerals (Guillot et al., 2015), a broad estimate of the confining pressure during shearing can be obtained by

assuming a geothermal gradient in the range of 20-35 °C/km. This geotherm is based on relatively high mantle temperatures in the Oligocene, as well as indications of a fairly thin lithosphere (Scott et al., 2014). Adopting this geotherm and a temperature of shearing in the range of 300-350 °C suggests a confining pressure of 270-400 MPa (roughly equivalent to 10-15 km depth). Sub-micron Raman spectroscopy mapping (Fig. 11b) reveals evidence for early-formed antigorite. This form of serpentine is generally thought to be stable at temperatures of >350 °C (Evans, 2004), suggesting that the shear zone experienced a

relatively early, higher-temperature deformation event(s). One possibility is that the early higher-temperature history may relate to deformation within a shear zone associated with ophiolite obduction (Harper et al., 1996; Hermann et al., 2000). However, because evidence for this early event was largely overprinted during development of the current steeply-dipping fabrics at greenschist-facies conditions, further work will be required to properly understand the significance of the antigorite porphyroclasts preserved in the Livingstone Fault.



## 5.2 A model for the internal structure and composition of plate boundary-scale serpentinite shear zones

Observations from eleven localities along a strike length of 140 km (Fig. 2) indicate that the Livingstone Fault contains a number of characteristic structural and compositional elements, which can be used to propose a general conceptual model for the structure of plate boundary-scale serpentinite shear zones (Fig. 12), drawing on features that have been described previously

in the literature (e.g., Bebout, 2013; Hirth and Guillot, 2013; Guillot et al., 2015; Tarling et al., 2018b, IN REVISION).

On a regional scale, the thickness of the shear varies between 5 m – 480 m (Figs. 2, 12a). Wall rocks belonging to the DMOB are variably serpentinised for distances of up to several hundreds of metres from the western shear zone margin (Fig. 12a). At Serpentine Saddle, the shear zone is particularly thick (420 m) and the ultramafic portions (serpentinised peridotite, massive serpentinite) of the DMOB wall rocks are absent (as at Four Brothers Pass and Beresford Pass). This suggests that the

ultramafic portions of the DMOB at these three localities may have been entirely converted in to scaly shear zone serpentinite (Fig. 12a). Quartzofeldspathic schists in the Caples-Aspiring wall rocks are cut by brittle faults that surround large lenses of metasomatised and fractured schist, which are ultimately plucked off and incorporated in to the shear zone (Fig. 12a).

The most important structural elements and processes within the Livingstone Fault are as follows:

(1) Scaly matrix serpentinite containing a pervasive foliation that wraps around pods (Fig. 12b-i). Although there is substan-

tial local dispersion in fabric orientations, on a regional scale the fabrics consistently indicate a steep, east-side up shear sense (Figs. 2, 7), consistent with previous work at West Burn (Craw, 1979) and in the Central Southland region (?Cawood, 1986).

(2) Large domains of massive serpentinite that preserve relatively intact and flat-lying rodingite dykes (Fig. 12b-ii). These areas are interpreted as 'low-strain domains' within the shear zone and likely preserve the initial dyke configurations prior to their entrainment and boudinage within the scaly matrix.

(3) Competent pods of massive serpentinite, veined serpentinite, rodingite, and schist that are entrained within the shear zone and cut by networks of brittle faults and fractures (Fig. 12b-iii). Brittle faults that cut isolated pods can have the same orientation as "C" shear bands in the surrounding scaly matrix (Fig. 12b-iii), suggesting geometric and kinematic linkage between distributed deformation in the matrix and more localised deformation in the pods. Where pods are in direct contact with each another, brittle fracture networks radiate outwards from the contact zone (Fig. 12b-iii), suggesting that local stresses in the

pods were transiently elevated as they were brought in to contact (Webber et al., 2018).

(4) Metasomatic reactions that occurred wherever serpentinites and schist are in contact, which includes the eastern boundary of the shear zone and the margins of schist pods (Fig. 12b-iv; Tarling et al., IN REVISION). Reactions between serpentinite and partially rodingitised gabbro/dolerite pods lead to the development of extensive metasomatic vein networks, which subsequently formed indurated pods of veined serpentinite. Additionally, reactions occurred between rodingite pods and the

surrounding scaly serpentinite, but only in cases where the continuity of the enveloping blackwall rim was disrupted by brittle faulting or shearing.

(5) Fault surfaces coated by layers of magnetite, which cut across the scaly fabric (Fig. 12b-v). These fault surfaces were described by Tarling et al. (2018b) who presented evidence for extremely localised dehydration of serpentinite found as inclusions within the magnetite layers. Tarling et al. (2018b) used numerical modelling to suggest that the localised dehydration





occurred by frictional heating during an ancient earthquake(s) with a magnitude between 2.7 and 4 (Fig. 12b-v).

### 5.3 Mixed rheology in plate boundary-scale serpentinite shear zones

The overall structure and composition of the Livingstone Fault may be representative of other large serpentinite-dominated
shear zones. In particular, the scale and composition of the shear zone, "block-in-matrix" style of deformation, and juxtaposition of ultramafic or mafic wall rocks against quartzofeldspathic wall rocks, are similar to the likely characteristics of the plate boundary-scale shear zone thought to be present along the slab-mantle interface in the shallow forearc region of subduction zones (Bebout and Barton, 1993, 2002; Fagereng et al., 2011; Bebout and Penniston-Dorland, 2016). The conditions under which deformation occurred in the Livingstone Fault are likely to be representative of lower pressures and possibly lower-
temperatures with respect to those at the slab-mantle interface in most subduction zones. However, the interpretations reached here regarding the structure of the shear zone, the potential importance of pressure-solution, and the influence of metasomatic reactions, are applicable over a relatively wide range of P-T conditions, suggesting that our observations are relevant to large serpentinite-bearing shear zones in different tectonic settings, including the slab-mantle interface.

Concentrations of magnetite along scaly foliation surfaces are interpreted to reflect preferential dissolution of serpentine
during deformation, and subsequent enrichment of magnetite along dissolution surfaces (Figs. 9,10, Rooney et al., 2018). Porphyroclasts of antigorite are elongate and show evidence for truncation by dissolution along their foliation-parallel margins, accompanied by the growth of "beards" of fibrous chrysotile (Fig. 9b). Additionally, abundant veins of fibrous chrysotile are found throughout the shear zone, either at the margins of individual serpentinite phacoids in the scaly fabric, or as slickenfibre veins along "C" shear bands and fractures cutting resistant pods. Collectively, these microstructures indicate that dissolution-
precipitation (or "pressure-solution") processes were important in forming the currently-preserved scaly serpentinite matrix (Imber et al., 1997; Jefferies et al., 2006; Wallis et al., 2015; Wintsch and Yi, 2002). This conclusion is consistent with previous work on dissolution-precipitation processes in serpentinite under greenschist or sub-greenschist facies conditions (Andréani et al., 2005, 2004).

Fracture networks radiating away from the contact areas between large pods suggest elevated stresses during interactions and
collisions between pods. This could reflect a 'log-jam' scenario (Fagereng and Sibson, 2010) where transient collisions in the shear zone occur episodically as pods are brought closer together. In regions of the shear zone where the pod content is relatively high compared to the matrix, these interactions may have transiently 'locked-up' regions of the shear zone, allowing stresses to accumulate and brittle failure to occur within the competent pods. Additionally, the polished and cataclastic margins of rodingite pods suggest that localised brittle faulting can also occur without direct physical interactions between pods. Numerical
modelling by Beall et al. (2019) shows that this can occur due to the development of elevated stresses around the margins of pods that are partially coupled to a surrounding ductiley deforming matrix.

Metasomatic reactions between serpentinite, schist and rodingite resulted in the transformation of foliated serpentinite in to a highly-indurated metasomatic fault rock consisting of networks of tremolite veins (Tarling et al., IN REVISION). These metasomatic reactions drove significant changes in fault rheology by generating abundant fluids, resulting in local fluid over-





pressure and ultimately to hydrofracturing of the foliated serpentinite shear zone and schist wall rocks (Tarling et al., IN REVISION). The close association of metasomatically-derived vein networks with brittle, cataclastic fault surfaces suggests that metasomatic reactions and the resultant hardening effects could promote a rheological transition from distributed ductile creep towards brittle deformation and localised slip (Tarling et al., IN REVISION).

Finally, we speculate that progressively concentrating magnetite along foliation surfaces and shear bands via a pressure-solution mechanism (Figs. 9,10) can ultimately lead to the development of continuous, through-going layers of magnetite that form important mechanical boundaries within the shear zone. Although the magnetite layers are thin (<1 mm) and represent a volumetrically minor (usually <5 wt. %, but up to 15 wt%) component of the scaly serpentinite, they provide interfaces that could potentially contribute to the localisation of strain, and perhaps the nucleation and/or propagation of dynamic instabilities.

Evidence presented by Tarling et al. (2018b) suggests that some magnetite layers can develop in to discrete fault surfaces several tens of metres long, which preserve evidence for transient dynamic rupture (Tarling et al., 2018b). The textural evolution from initially-disseminated magnetite towards continuous magnetite-coated fault surfaces is a poorly constrained process, but this could be a relevant question to address using a combination of field, microstructural and experimental observations.

## 6 Conclusions

The Livingstone Fault is a plate boundary-scale serpentinite shear zone that is tens to several hundreds of metres wide. The bulk of the shear zone consists of a pervasive scaly fabric dominated by chrysotile and lizardite/polygonal serpentine. The scaly fabric wraps around fractured and faulted pods of massive serpentinite, rodingite and partially metasomatised quartzofeldspathic schist up to hundreds of metres long. Well-developed S-C fabrics in the scaly serpentinite indicate an east-side up shear sense, and preserve textural evidence to suggest that pressure solution was an important deformation mechanism during shearing.

Metasomatic reactions were ubiquitous wherever serpentinite contacted schist or rodingite, forming multi-generational vein networks filled by nephritic tremolite. Based on field and microstructural observations, we present a conceptual model of the structure and composition of large serpentinite shear zones deforming at greenschist facies conditions. The model involves bulk distributed deformation by pressure-solution creep, accompanied by localised brittle deformation within pods or along magnetite-coated fault surfaces. Metasomatic reactions can generate in-situ fluid overpressures that may trigger hydrofractur-

ing and the formation of vein networks, leading to reaction hardening and embrittlement within metasomatised portions of the shear zone. The scale, internal structure, and composition of the Livingstone Fault and its wall rocks suggest that it could provide a suitable analogue for other plate boundary-scale serpentinite shear zones, including the serpentinite-bearing shear zone expected to occur along the slab-mantle interface in subduction zones.

*Author contributions.* M.S.T, S.A.F.S and J.M.S. carried out fieldwork and performed microstructural analysis of fault rocks. M.S.T and

C.V performed Transmission Electron Microscopy. M.S.T. and J.S.R performed Raman analysis with input from K.C.G. M.S.T wrote the manuscript with discussion and input from all authors. S.A.F.S and J.M.S supervised the project.





*Acknowledgements.* This work was supported by the Marsden Fund Council (project UOO1417 to Smith) administered by the Royal Society Te Apārangi, with additional funding from a University of Otago Research Grant. Luke Easterbrook provided valuable assistance in drone imagery and photogrammetry. Chris Tulley provided field assistance on many of the field expeditions and contributed data on the Mount Raddle section collected during his BSc Hons project. Jordan Crase contributed data on the Serpentine Saddle, Cow Saddle, Fiery Col and

5 Cosy Gully sections collected during his MSc. project. We thank Marianne Negrini, Brent Pooley, Claudia Magrini and Giovanna Giorgetti for technical support.





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



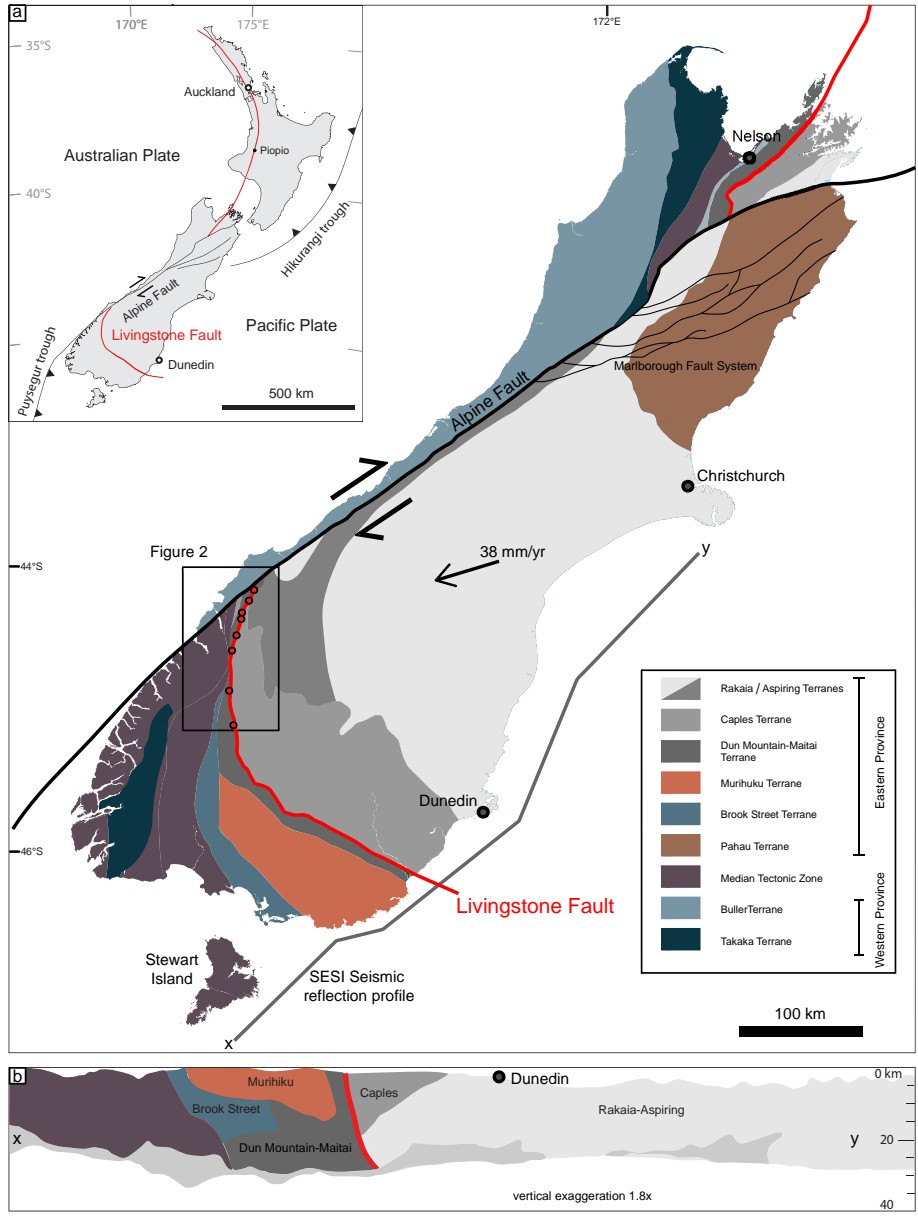

**Figure 1.** Regional geological setting. a) Simplified regional map of the basement geology and tectonic setting of the South Island of New Zealand. Inset shows a map of New Zealand with present-day plate boundaries and location of the Livingstone Fault in the North and South Islands. Piopio (inset map) is the only exposure of the Dun Mountain Ophiolite Belt in the North Island of New Zealand (O'Brien and Rodgers, 1973). Modified from Cooper and Ireland (2015) with data from Mortimer (2004). (b) Regional-scale cross section along line x-y (shown in part a) based on interpretation of the composite South East South Island (SESI) seismic reflection profile by Mortimer et al. (2002).



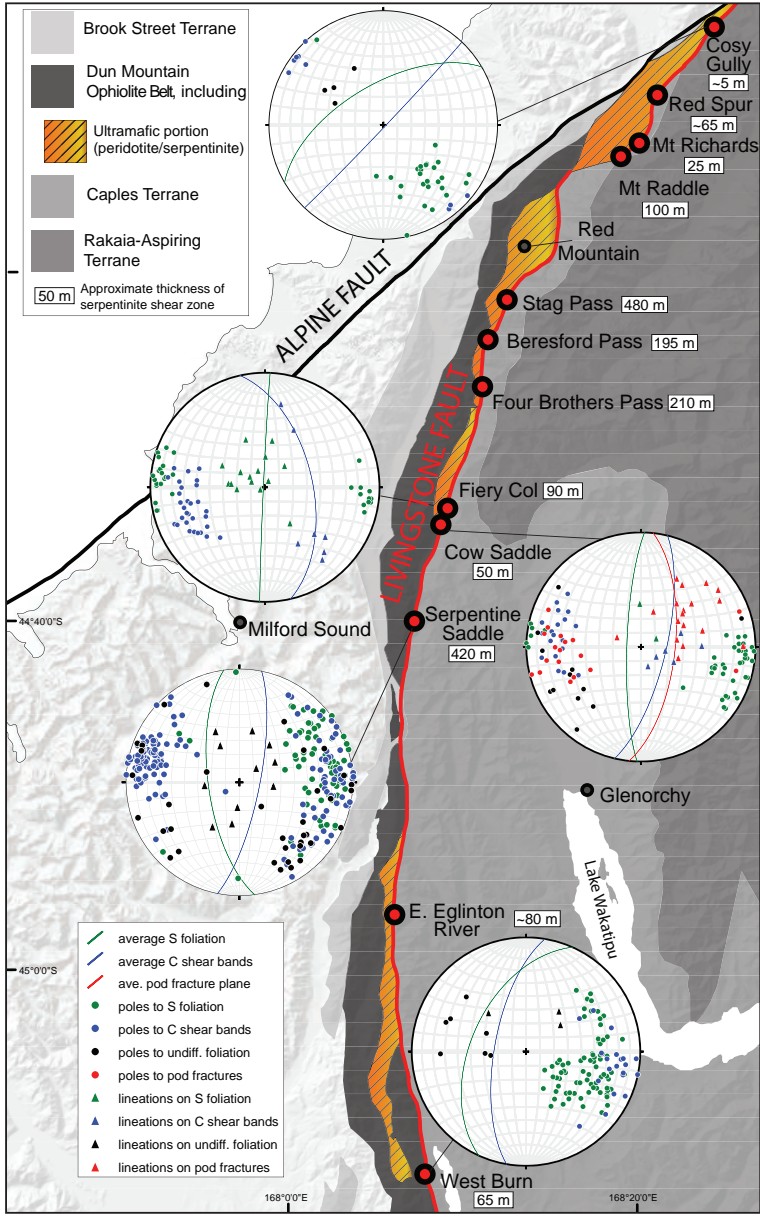

**Figure 2.** Simplified geological map of the Livingstone Fault with eleven study sites marked, from Cosy Gully in the north to West Burn in the south. Approximate thickness of the serpentinite shear zone based on field observations is shown at each locality. Stereonets (all lower hemisphere, equal area) represent measurements of the main shear zone fabrics, including undifferentiated scaly foliations, S-C fabrics where present, and lineations. At Cow Saddle, additional data are provided for the orientations of brittle faults and associated lineations that cut through pods of massive serpentinite. Stereonets produced using Stereonet v. 10.1.6 (Allmendinger et al., 2011)).




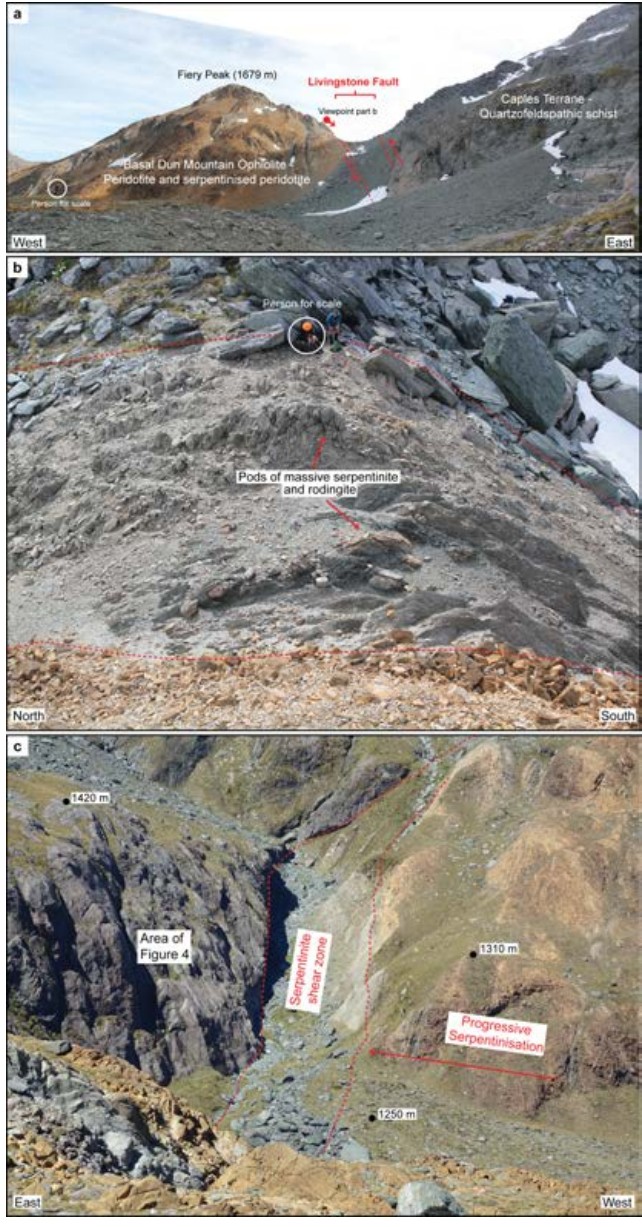

**Figure 3.** Typical field exposures of the Livingstone Fault where it crosses high passes. Red dashed lines approximate the boundaries of the serpentinite shear zone. (a) Panorama of Fiery Col viewed from the south. Person for scale circled in white. Here, the shear zone separates massive serpentinite and serpentinised peridotite from schists of the Caples Terrane. (b) Fiery Col viewed from below Fiery Peak (location shown in part a). Elongate pods of massive serpentinite are surrounded by scaly serpentinite. The boundaries between the serpentinite shear zone and the wall rocks are well-defined. (c) Cow Saddle viewed from the north. The DMOB wall rocks are progressively serpentinised towards the western boundary of the shear zone. The area shown in Figure 4 is also indicated.



**Figure 4.** Deformation in the Caples Terrane wall rocks at Fiery col and Cow Saddle. (a) Photo looking south from Fiery Col towards Cow Saddle (location shown in Figure 3c). Schists adjacent to the eastern boundary of the serpentinite shear zone are cut by linked arrays of brittle faults that surround large lenses of metasomatised and fractured schist. Tent for scale circled in white. (b) The brittle faults are characterised by polished slickensides with lineations, and cataclastic slip zones. Stereonet (lower hemisphere, equal area) shows orientations of eight fault surfaces with associated steeply-plunging lineations.



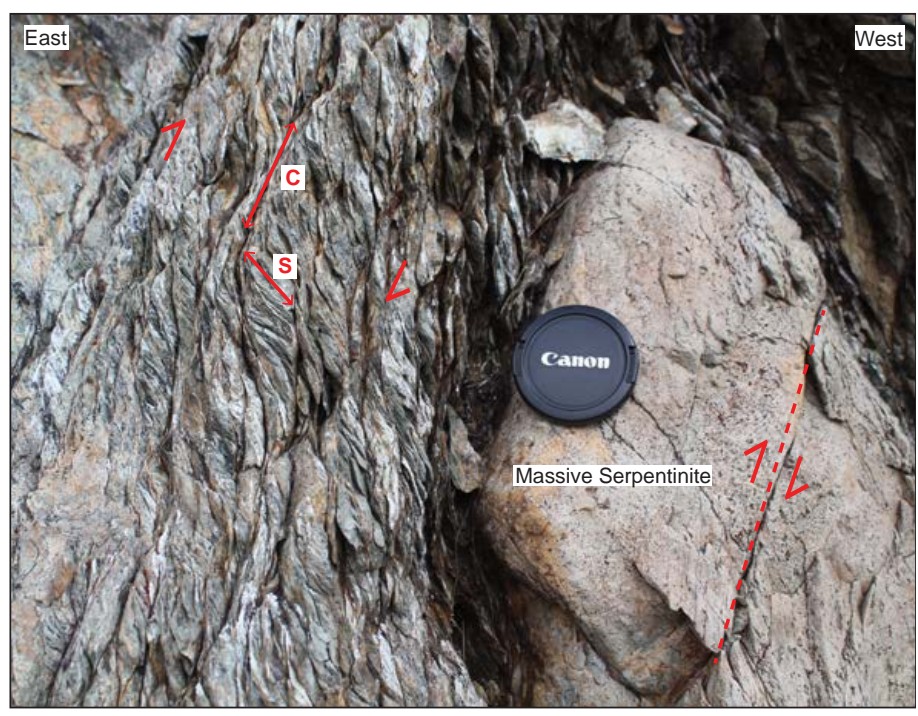

**Figure 5.** Photo showing well-defined S-C fabrics at Cow Saddle, wrapping around a fractured pod of massive serpentinite. Structural data from Cow Saddle are shown in Figure 2. The asymmetry of the S-C fabric, combined with moderately- to steeply-plunging lineations on foliation surfaces and shear bands, suggests an east-side up (i.e. Caples-side up) shear sense. Fractures that cut through pods of massive serpentinite at this locality are subparallel to the C shear bands in the surrounding scaly matrix, and have small offsets consistent with an east side-up shear sense.



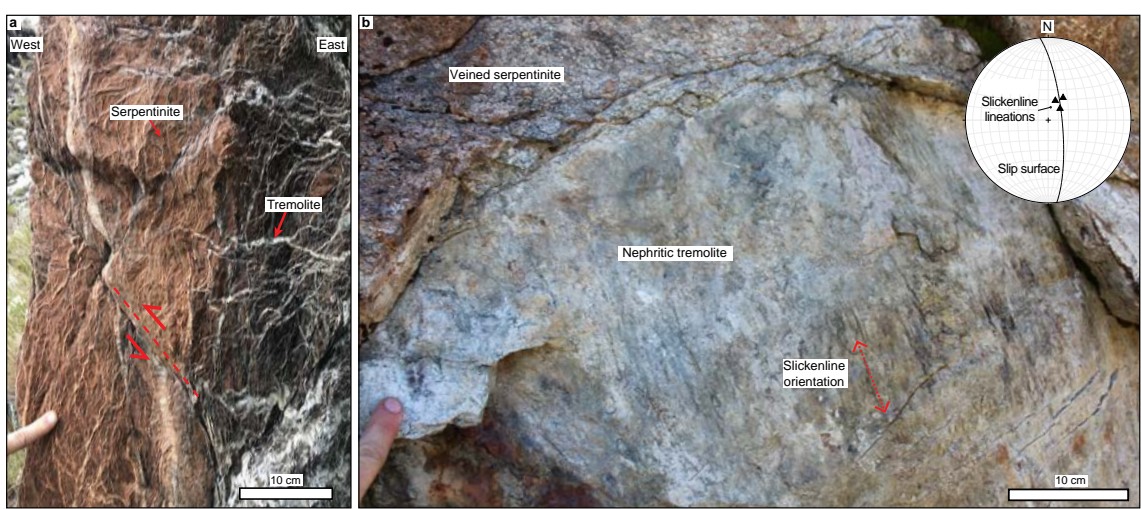

**Figure 6.** Tremolite vein networks and associated discrete slip surfaces. (a) Multigenerational tremolite vein network at the contact between serpentinite and Caples Terrane schist. Cross-cutting fault noted in red. (b)A discrete slip surface cross-cutting the metasomatic reaction zone between serpentinite and Caples Terrane schist displays steeply plunging slickenlines consistent with overall fault kinematics.



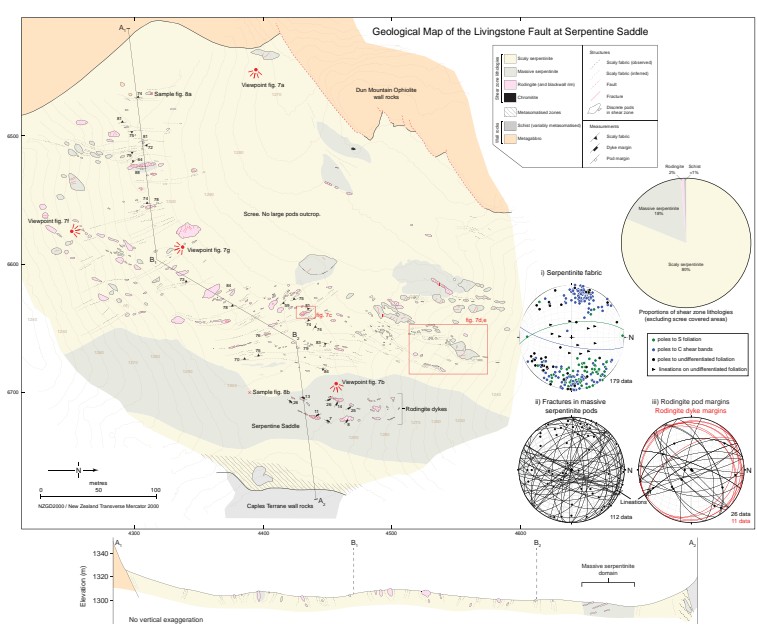

**Figure 7.** Detailed geological map and cross-section of the Livingstone Fault at Serpentine Saddle derived from field mapping on to high-resolution, drone-acquired orthophotos. The map highlights the internal structure and composition of the serpentinite shear zone at this locality, including the traces of the scaly fabrics, pods of rodingite and massive serpentinite, and larger domains of massive serpentinite containing gently-dipping rodingite dykes. The location of images shown in Figure 8, and some samples shown in Figure 9, are also indicated. Stereonets (lower hemisphere, equal area) show measurements of (i) S-C fabrics, undifferentiated scaly foliation, and lineations in the shear zone, (ii) brittle faults and associated lineations that cut through massive serpentinite pods, and (iii) the margins of rodingite dykes and pods, and lineations on slickensided margins of pods. Pie chart shows the abundance of the most important lithologies within the shear zone in the mapped area (wall rocks are excluded). Light brown lines are topographic contours at 5 metre intervals.



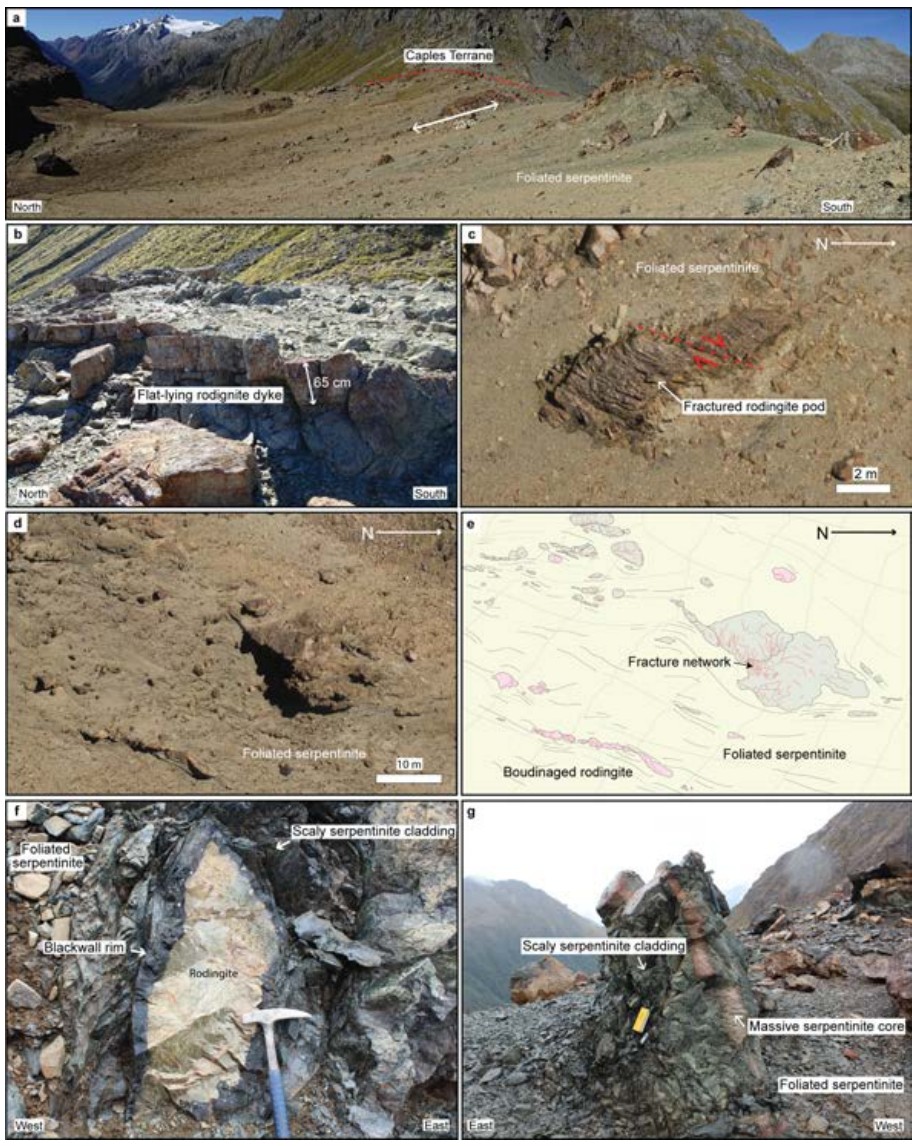

**Figure 8.** Shear zone structures at Serpentine Saddle. Location of images shown on the map in Figure 7. (a) Panorama of Serpentine Saddle looking east towards the Caples Terrane wall rocks. Upstanding spines on the ridgeline are elongate pods of massive serpentinite and rodingite. (b) Gently-dipping rodingite dykes in a large massive serpentinite domain. (c) High-resolution orthophoto showing intensely fractured and boudinaged rodingite pod surrounded by scaly serpentinite. (d) High-resolution orthophoto and (e) corresponding line tracing showing a boudinaged rodingite dyke (bottom left in both figures) and two large pods of massive serpentinite that have been brought in to contact. A dense network of fractures radiates outwards from the contact region between the two pods of serpentinite. (f) Rodingite pod enclosed by a blackwall rim and surrounding matrix of scaly serpentinite. (g) Serpentinite pod with a massive serpentinite core and a cladding that transitions to scaly serpentinite.





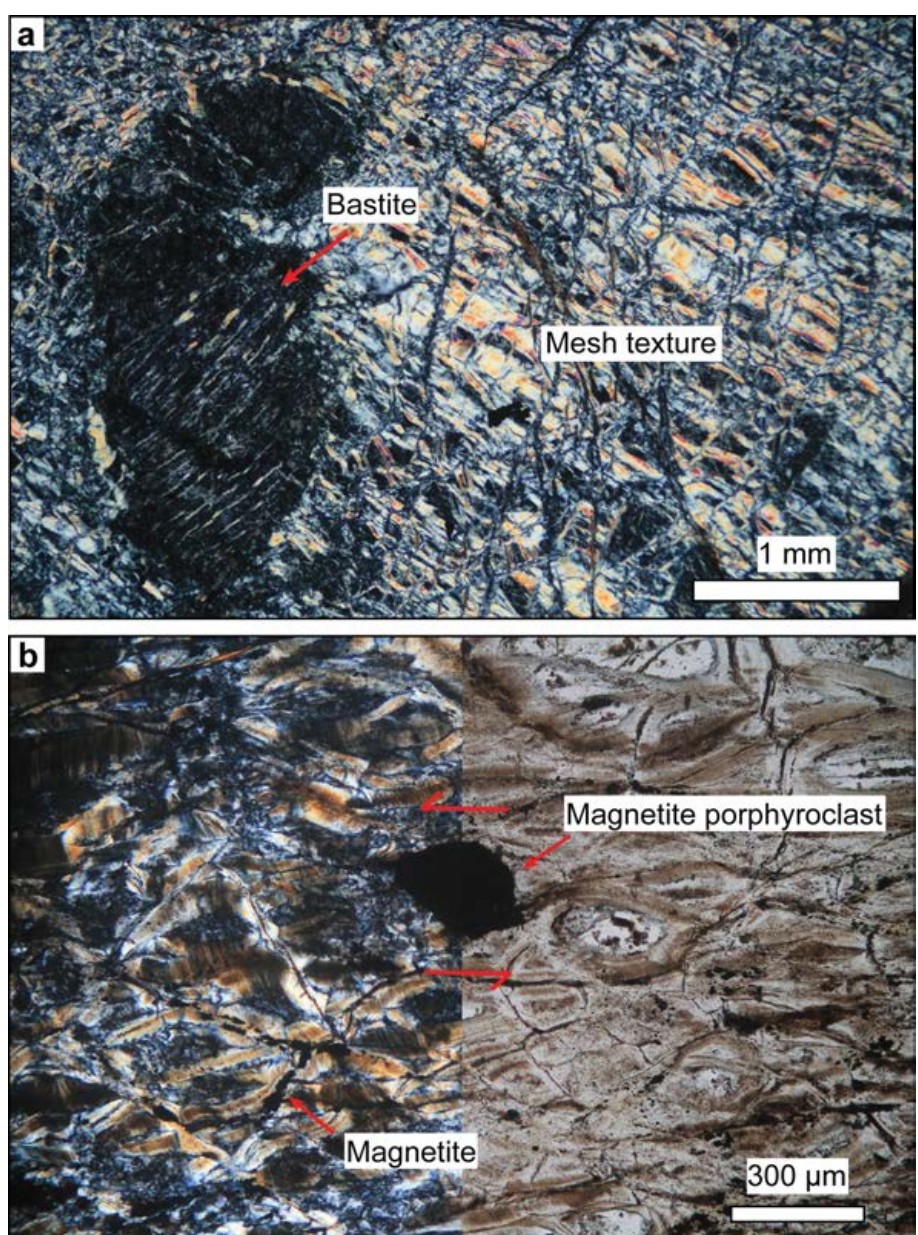

**Figure 9.** Serpentinite textures in pods and massive serpentinite domains. (a) Mesh textured and bastite serpentinite from the centre of a serpentinite pod (location in Fig/ 7). Optical microscope image in crossed-polarised light. (b) Sheared mesh textured serpentinite in the initial stages of developing into 'ribbon serpentinite' from within the large domain of massive serpentinite at Serpentine Saddle (location in Fig. 7) Optical microscope image in crossed-polarised light (left hand side) and plane polarised light (right hand side).



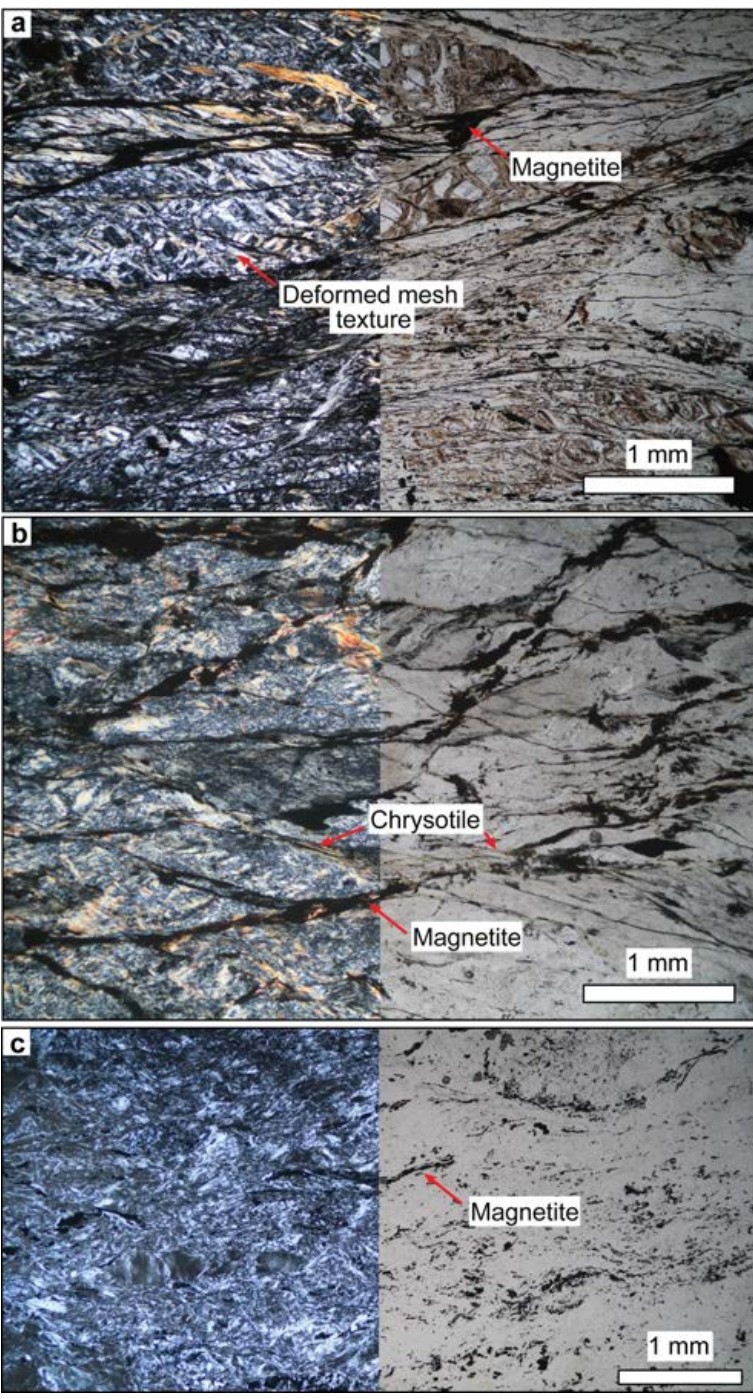

**Figure 10.** Evolution of serpentinite texture and mineralogy in scaly shear zone serpentinites. Each figure shows an optical microscope image in crossed-polarised light (left hand side) and plane polarised light (right hand side). (a) Where the scaly foliation is relatively widely spaced, magnetite-bound phacoids preserve weakly deformed mesh textured serpentinite. (b) Where the scaly foliation is more closely spaced, no recognisable pseudomorphic textures are preserved. Continuous seams of magnetite outline the scaly foliation. (c) In samples of well-developed scaly serpentinite with lower proportions of magnetite, discontinuous seams of magnetite are concentrated at the boundaries of the scaly foliation, broadly defining lenticular domains of serpentinite.



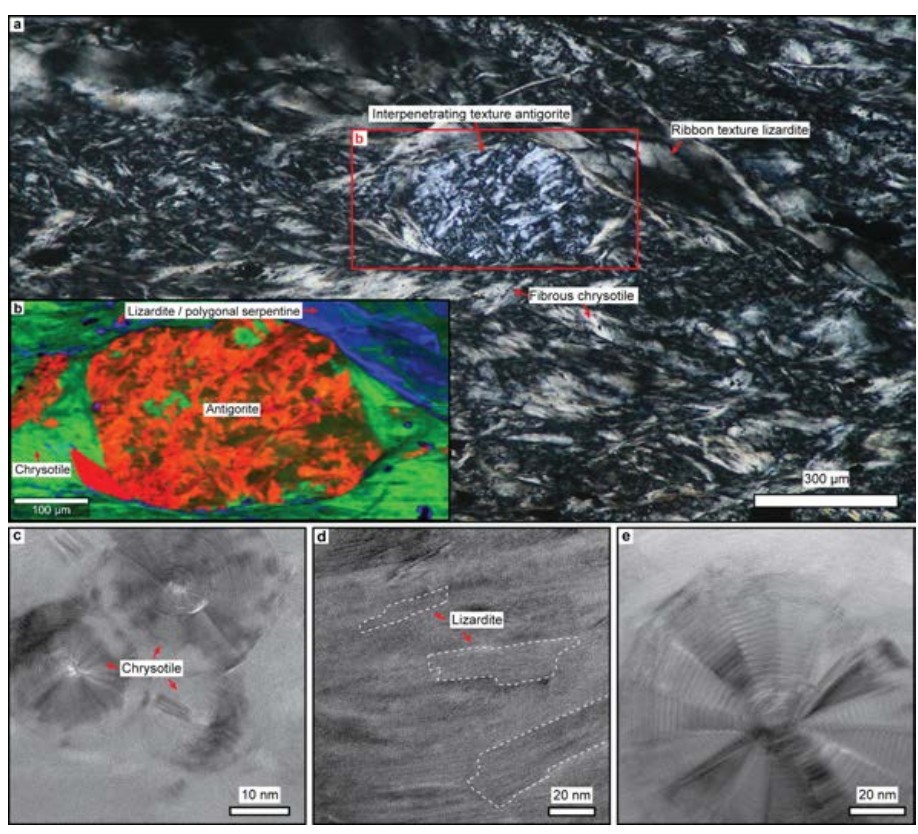

**Figure 11.** Raman and TEM characterisation of the scaly serpentinite fabric. (a) Representative thin section photomicrograph (crossed-polarised) of scaly serpentinite composed dominantly of chrysotile with ribbon-textured lizardite, with a porphyroclastic lens of antigorite showing an interpenetrating texture. (b) Submicron Raman spectroscopy map of the antigorite porphyroclast shown in part a. The map highlights a central core of antigorite (red) that is breaking down to chrysotile (green) and lizardite/polygonal serpentine (blue). The porphyroclast is elongate: margins subparallel to the scaly foliation are truncated by lizardite/polygonal serpentine-rich seams or aggregates, whereas fibrous chrysotile grows in "beards" from the ends of the porphyroclast. (c) TEM image of chrysotile fibre cross-sections (d) TEM image of laths of lizardite (e) TEM image of a polygonal serpentine cross-section.

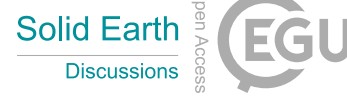

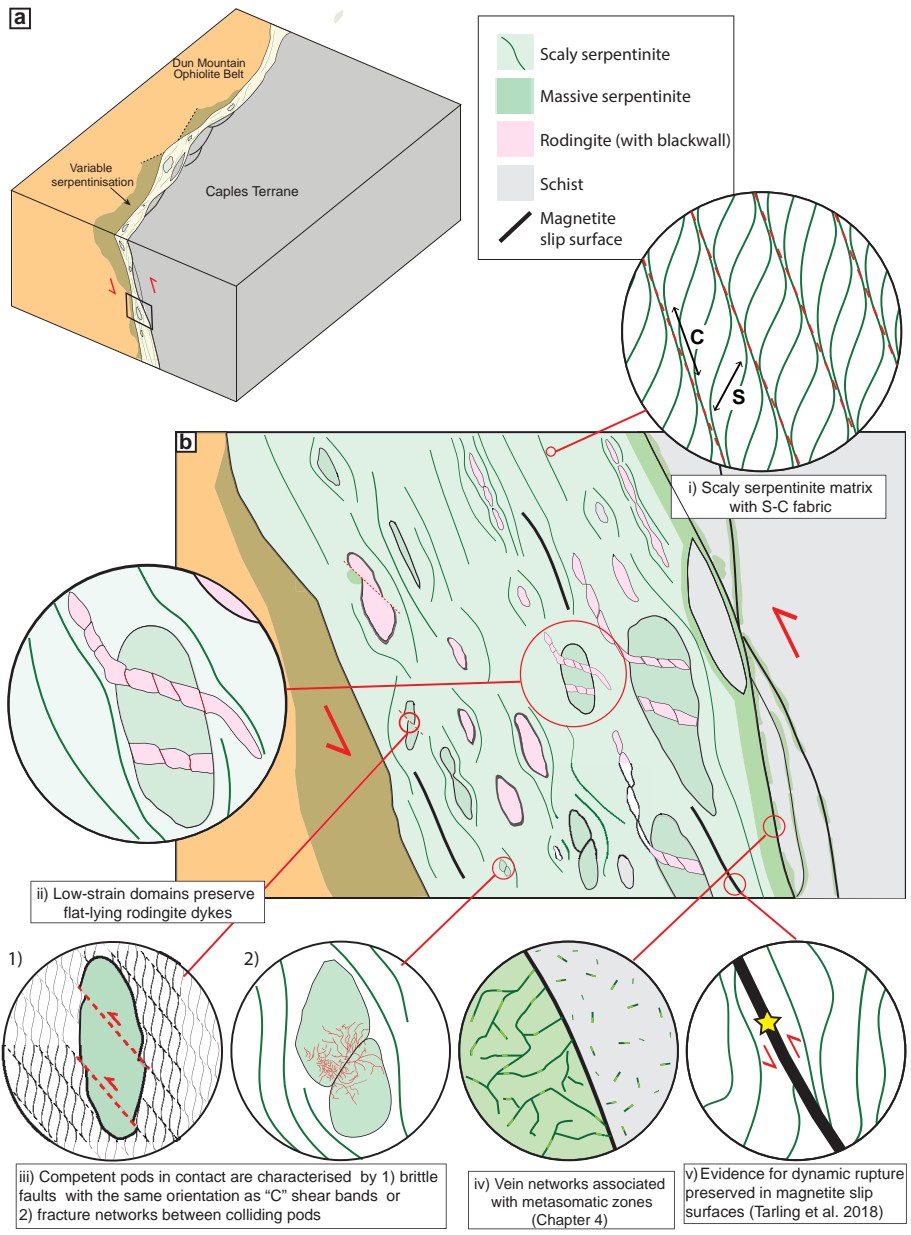

**Figure 12.** A conceptual model of the structure and composition of large serpentinite shear zones, based on the most important structural elements and deformation processes observed in the Livingstone Fault. (a) Regional-scale characteristics. The thickness of the Livingstone Fault, and the degree of serpentinisation in the DMOB wall rocks, are highly variable along strike. The Caples/Aspiring wall rocks are cut by networks of brittle faults that enclose large lenses of fractured and metasomatised schist. (b) Internal structure and composition. i) Scaly matrix serpentinite with a pervasive, steeply-dipping foliation ii) 'Low-strain domains' of massive serpentinite preserve flat-lying rodingite dykes that are sheared in to the surrounding scaly matrix. iii) Pods contain 1) brittle faults with a similar orientation to the "C" shear bands, or 2) fracture networks that radiate outwards from contact regions between colliding pods. iv) metasomatic reactions led to the development of vein networks at contact between serpentinite and the Caples Terrane. v) Discrete magnetite-coated slip surfaces preserve evidence for dynamic rupture and coseismic dehydration of serpentinite (Tarling et al., 2018b).