# Peer review of "The internal structure and composition of a plate boundary-scale serpentinite shear zone: The Livingstone Fault, New Zealand"

_Solid Earth, 2019_

## Referee Comment (RC1) · Anonymous Referee #1 · 18 Apr 2019

Review of Tarling et al.'s Solid Earth manuscript titled "The internal structure and composition of a plate boundary-scale serpentinite shear zone: The Livingstone Fault, New Zealand"

General comments (overall quality)

This excellent manuscript documents the complex macro- and micro-structures of the Livingstone Fault, a >1000 km long serpentinite shear zone exposed on the South Island of New Zealand. The mechanical behaviour of serpentinite likely plays a major role in controlling the rheology of major faults, including plate boundaries. In particular, serpentinite is very likely to be present along subduction-zone plate boundaries where

the subducting plate descends beneath the forearc mantle. The extent to which a subducting plate induces flow in the mantle wedge may depend in part on the rheologic coupling across a serpentinite shear zone. This important study provides important insight into how serpentinite deforms at elevated temperature and pressures, and on the scale of tens to hundreds of metres. This is a very well-written paper that strongly merits publication in Solid Earth after considering my minor comments below. I note that the paper is exceptionally well illustrated with clearly annotated field photos and photomicrographs. The field photos are spectacular!

Specific comments (individual scientific questions/issues)

Page 2 (Introduction) – I recommend the authors add a sentence or two providing specific examples of other serpentinite-bearing shear zones around the world.

Page 9 (sections 4.3.1), Figures 9 and 10, and throughout the text – I recommend the authors specify, to the extent possible, the type(s) of serpentine minerals present in the different serpentinites. Reading between the lines, the massive serpentinite described in section 4.3.1 is likely composed of chrysotile + lizardite (+ magnetite); antigorite and other forms of serpentine are rare.

Page 10, line 15-16. I recommend expanding the first sentence "... with an estimated ambient temperature during shearing of 300-350 °C" to articulate the constraints on the estimated temperatures, citing appropriate references. The assemblage lizardite + chrysotile can occur over a broader temperature range, but perhaps the general absence of brucite and antigorite is being used to narrow the temperature estimate.

Page 13, line 24 – What is the evidence that metasomatic reactions "can generate in-situ fluid overpressures?" Would this not depend on the specific metasomatic reaction (e.g., cation exchange vs. dehydration reaction)?

Technical corrections (typos, etc.)

Page 3, line 5 and 14 – The rocks between the Western and Eastern Provinces are
referred to as the Median "Batholith" in the text and the Median "Tectonic Zone" in Figure 1. I recommend using one or the other for consistency.

Page 2, line 23; page 9, line 20; page 11, line 16 - The question marks in cited references should be deleted and appropriate references added where necessary.

Page 21, Figure 1 – Several of the colours chosen for the geologic map are very similar making it difficult to distinguish some of the units (Dun Mountain terrane versus Median Tectonic Zone) particularly when similar coloured units are juxtaposed (e.g., contact between Dun Mountain terrane and Brook Street terrane). I recommend selecting different colours to ensure the geologic map can be easily interpreted.

Page 32, Figure 12 – "(Chapter 4)" should be removed from box (iv)

Page 27, Figure 7 – This detailed geologic map should be published at the largest size possible (i.e., full page width). At its current size most of the details are not legible.

---

## Referee Comment (RC2) · Telemaco Tesei (Referee) · 10 May 2019

Tarling and co-workers present an excellent description of the Livingstone Fault in the South Island of New Zealand. This manuscript is very interesting because it presents data and observations from outcrops along a strike of several tens of kilometers, which is not common. Albeit most of the work focuses on a single outcrop, it is nevertheless important to document the lateral continuity and structure of fault zones that can be highly heterogeneous. These informations on fault structure and composition are presented in the context of nearby terranes and are presented in a effective way. Outcrop-scale observations are then well integrated with micro and nano-structural analysis to

form a nice global picture of a large serpentine-bearing fault. The mechanics of scaly serpentinite faults presented by Tarling et al. is consistent with previous literature on the subject but also highlights some structural complexity of the fault zone. In particular, the documentation of processes at the contact with the wall rocks of different composition and the potential rheological influence of non-serpentinitic blocks within the fault core are important evidence brought about in this manuscript. I think the manuscript needs only very minor revisions before being accepted for publication.

Telemaco Tesei Durham University

Comments: I reckon that references should be in chronological. Labels within field figures are all very small and difficult to read.

Page 2, line 16 and following. There are several more studies that document with some detail the processes and the structure of serpentine-bearing fault zones. Some potential additions to the list: Maltman, 1978; Williams, 1979; Twiss and Gefell, 1990; Alexander & Harper, 1992; Gates, 1992; Bailey et al., 2000, Hirauchi and Yamaguchi, 2007, Bellot, 2008, Melosh, 2019. In the following lines: it might be worth mentioning with some more details what these studies say about the structure and deformation of serpentinite-bearing faults. Limiting the discussion to the characteristic scaly fabric is a bit over simplistic. In the discussion, it might be useful to highlight the differences with the previous knowledge about serpentinite faults.

Page 8, line 30: there is a question mark after "Vannucchi et al., 2003" what does it means?

Page 9, line 19. See also Melosh, 2019, G3

Page 10, line 4: How did you identify these minerals? how large are the grains?

Page 10; line 16: The temperature range in which chrysotile and lizardite are stable is much wider than 300-350°C. The absence, or instability, of antigorite may well set the high temperature boundary, but not the lower boundary. Since two close terranes

have zeolite facies (T<200°C) and Prehnite-Pumpellyte facies (T<300°C) metamorphic imprint, it is possible that the Livingstone fault was active at temperatures lower than 300°C.

Page 11, line 5: I think a paper in revision does not qualifies as previous literature. line 8-10: this is a very interesting observation. Do you have an estimation of the thickness of the mantle section in the other outcrops of the ophiolite adjacent to the Livingstone fault? I would be interesting to understand this change in thickness. For instance, are there any changes in kinematics or amount of displacement that could account for this change in thickness of the deformed mantle portion?

Page 12, line 12: "...P-T conditions". Maybe a reference or two are necessary here. line 21. from the sentence it looks like that the references talk about pressure solution producing the scaly fabric of serpentinites. They are only some examples of work about pressure-solution weakening of faults in general. Maybe add "similarly to what happens in other faults"? or something similar. line 34: it might be worth to metionting that the association of serpentinites /ultramafics with tremolite (and chlorite and talc) is well documented in the literature (e.g. Cronshaw 1923; Nishiyama 1990; Boschi et al., 2006; Bach and Klein 2009 among many others). In particular at the contact with different lithologies (rodingites but also metasediments).

Page 13, line 24. This statement about overpressure induced by metasomatic reactions is a bit vague. I would suggest to either remove it or present the evidence for such a phenomena (I don't think Fig. 6a is enough).

---

## Editor Comment (EC1) · Cristiano Collettini (Editor) · 15 May 2019

Dear Authors, I have now received two reviews of your manuscript and both reviews emphasize the scientific excellence of your contribution. They also highlight some minor points that you can easily address during the revision round.

Sincerely Cristiano Collettini

---

## Author Comment (AC1) · 30 May 2019

We thank reviewer 1 for their constructive comments which will allow us to clarify and improve many aspects of the manuscript. We agree with all of the reviewers' comments and intend to make changes to the text and figures in response to each comment. Below we include a point-by-point response to the reviewers' comments outlining our intentions.

Reviewer comments are in **bold**, author responses are in *italics.*

[Figure]

Specific comments (individual scientific questions/issues)

**Page 2 (Introduction) – I recommend the authors add a sentence or two providing specific examples of other serpentinite-bearing shear zones around the world.**

*As recommended, we will expand the introduction to include several other specific examples of previously-reported serpentinite-bearing shear zones around the world. Additionally, in response to comments by Reviewer Telemaco Tesei, we will expand our brief overview of serpentinite-bearing shear zones and broaden the reference list referring to previous studies of serpentinite shear zones, including references by Maltman, 1978; Williams, 1979; Twiss and Gefell, 1990; Alexander Harper, 1992; Gates, 1992; Bailey et al., 2000, Hirauchi and Yamaguchi, 2007, Bellot, 2008, Melosh, 2019.*

**Page 9 (sections 4.3.1), Figures 9 and 10, and throughout the text – I recommend the authors specify, to the extent possible, the type(s) of serpentine minerals present in the different serpentinites. Reading between the lines, the massive serpentinite described in section 4.3.1 is likely composed of chrysotile + lizardite (+ magnetite); antigorite and other forms of serpentine are rare.**

*We agree with the reviewer that we can be more precise in our definition of the different serpentine varieties, and we have put substantial effort in to developing new techniques of Raman Spectroscopy to characterise the serpentine minerals. Wherever possible in the updated manuscript, we will add details on the types of serpentine minerals present.*

**Page 10, line 15-16. I recommend expanding the first sentence ". . . with an**

**estimated ambient temperature during shearing of 300-350 C" to articulate the constraints on the estimated temperatures, citing appropriate references. The assemblage lizardite + chrysotile can occur over a broader temperature range, but perhaps the general absence of brucite and antigorite is being used to narrow the temperature estimate.**

*In response to this comment as well as a similar remark made by reviewer Telemaco Tesei, we will apply a more conservative stance with regards to the temperature range and broaden our initial estimates. We will expand our treatment of the evidence for the temperature estimates, including the instability of antigorite, the general lack of the assemblage antigorite + brucite, the metamorphic facies in the wall rocks and the dominance of a chrysotile + lizardite assemblage.*

**Page 13, line 24 – What is the evidence that metasomatic reactions "can generate insitu fluid overpressures?" Would this not depend on the specific metasomatic reaction (e.g., cation exchange vs. dehydration reaction)?**

*Yes, the reviewer is correct, and this is the subject of another paper focused on the metasomatic reactions. In the revised manuscript, we will revise this statement to remove any mention of fluid overpressure.*

Technical corrections (typos, etc.)

**Page 3, line 5 and 14 – The rocks between the Western and Eastern Provinces are referred to as the Median "Batholith" in the text and the Median "Tectonic Zone" in Figure 1. I recommend using one or the other for consistency.**

*We will revise Figure 1 such that the geological unit between Western and Eastern provinces is consistently referred to as the Median Batholith throughout the paper.*

**Page 2, line 23; page 9, line 20; page 11, line 16 - The question marks in cited references should be deleted and appropriate references added where necessary.**

*The errors in referencing will be addressed.*

**Page 21, Figure 1 – Several of the colours chosen for the geologic map are very similar making it difficult to distinguish some of the units (Dun Mountain terrane versus Median Tectonic Zone) particularly when similar coloured units are juxtaposed (e.g., contact between Dun Mountain terrane and Brook Street terrane). I recommend selecting different colours to ensure the geologic map can be easily interpreted.**

*Following this recommendation, the colour palette will be adjusted so that there is a greater contrast between adjacent units on the map.*

**Page 32, Figure 12 – "(Chapter 4)" should be removed from box (iv) Page 27, Figure 7 –**

*The erroneous label in Figure 12 box (iv) will be removed.*

**This detailed geologic map should be published at the largest size possible (i.e., full page width). At its current size most of the details are not legible.**

*We note that the geological map that was embedded in the submitted manuscript .pdf was very low-resolution, but that the supplementary file we submitted provided a high-resolution version. Our intention was that the detailed geologic map should be published at high resolution at A3 size in the .pdf version of the paper, and we will request this option with the journal editors and typesetters. In the online version of the paper, the map will be high resolution and zoomable, which will allow for all details to be clearly legible.*

---

## Author Comment (AC2) · 30 May 2019

We thank the reviewer for his detailed and constructive comments. We agree with all comments and intend to incorporate all the reviewers' suggestions into the revised manuscript. Below we include a point-by-point response to the reviewers' comments.

Reviewer comments are in **bold**, author responses are in *italics*.

[Figure]

**General comments: I reckon that references should be in chronological. Labels within field figures are all very small and difficult to read.**

*The references will be revised and changed to be ordered chronologically in the text. We will make sure that all labels and annotations within the figures are clearly legible, and we will increase the font point sizes where necessary to improve legibility. Unfortunately, some of the figures that were submitted in the original .pdf were converted to a lower resolution than we would have liked, which may have rendered some of the text difficult to read. We will ensure that final figure versions are submitted at highest resolution.*

Specific comments:

**Page 2, line 16 and following. There are several more studies that document with some detail the processes and the structure of serpentine-bearing fault zones. Some potential additions to the list: Maltman, 1978; Williams, 1979; Twiss and Gefell, 1990; Alexander Harper, 1992; Gates, 1992; Bailey et al., 2000, Hirauchi and Yamaguchi, 2007, Bellot, 2008, Melosh, 2019. In the following lines: it might be worth mentioning with some more details what these studies say about the structure and deformation of serpentinite-bearing faults. Limiting the discussion to the characteristic scaly fabric is a bit over simplistic. In the discussion, it might be useful to highlight the differences with the previous knowledge about serpentinite faults.**

*We will revise the introduction to expand the overview of serpentinite-bearing shear zones worldwide and include the suggested references. Additionally, in response to comments by Reviewer 1, we will also provide specific reference to several examples*

*of other serpentinite-bearing shear zones around the world.*

**Page 8, line 30: there is a question mark after "Vannucchi et al., 2003" what does it means?**

*Errors in referencing such as this question mark (and others throughout the text) will be addressed. The reference in question missing here is "Vannucchi 2019".*

**Page 9, line 19. See also Melosh, 2019, G3**

*References will be updated to include Melosh 2019.*

**Page 10, line 4: How did you identify these minerals? how large are the grains?**

*The minerals were identified with scanning electron microscopy energy-dispersive X-ray spectroscopy (SEM-EDS) and Raman Spectroscopy (where crystallographic symmetry permits; details in Tarling et al 2018; Rooney et al. 2018). Grains are typically tens of microns in size, but can be up to mm-scale. The text will be updated to include these details.*

**Page 10; line 16: The temperature range in which chrysotile and lizardite are stable is much wider than 300-350 C. The absence, or instability, of antigorite may well set the high temperature boundary, but not the lower boundary. Since two close terranes have zeolite facies (T<200 C) and Prehnite-Pumpellyte facies (T<300 C) metamorphic imprint, it is possible that the Livingstone fault was active at temperatures lower than 300 C.**
*Following the advice of the reviewer as well as a similar comment made by Reviewer 1, we will apply a more conservative stance with regards to the temperature range and broaden our initial estimates. We will expand our treatment of the evidence for the temperature estimates, including the instability of antigorite, the general lack of the assemblage antigorite + brucite, the metamorphic facies in the wall rocks and the dominance of a chrysotile + lizardite assemblage. Additionally, the general lack of incohesive brittle fault rocks in the serpentinite shear zone would argue against any significant very shallow, low temperature deformation. Overall, the lines of evidence are compatible with a temperature range of 250-350 C.*

**Page 11, line 5: I think a paper in revision does not qualifies as previous literature.**

*We will remove the paper in revision from the reference list.*

**Line 8-10: this is a very interesting observation. Do you have an estimation of the thickness of the mantle section in the other outcrops of the ophiolite adjacent to the Livingstone fault? I would be interesting to understand this change in thickness. For instance, are there any changes in kinematics or amount of displacement that could account for this change in thickness of the deformed mantle portion?**

*These are interesting questions raised by the reviewer. We have not mapped in detail the ultramafic sections of the ophiolite belt adjacent to the Livingstone Fault, although crude estimates of thickness can be obtained from satellite/aerial imagery and other field observations (including other published articles and maps). Based*

*on this, we have not noted any specific changes in shear zone kinematics that could be related to the thickness of the ultramafic portions of the ophiolite or the width of the serpentinite shear zone. However, we believe that this kind of correlation would require a more targeted project, focusing specifically on whether there is a correlation between shear zone structure/kinematics and the structure (e.g. thickness, degree of serpentinisation) of the wall rocks. These questions may form the basis for future field work. Unfortunately, we lack constraints on shear zone displacement due to the absence of clear offset markers or boundaries, and thus any correlations between displacement and e.g. kinematics/wall rock structure, are currently not possible to constrain.*

**Page 12, line 12: ": : :P-T conditions". Maybe a reference or two are necessary here.**

*We will add references to support the notion that the interpretations reached regarding the structure, importance of pressure-solution, and metasomatic reactions are relevant to a wide range of P-T conditions.*

**line 21. from the sentence it looks like that the references talk about pressure solution producing the scaly fabric of serpentinites. They are only some examples of work about pressure-solution weakening of faults in general. Maybe add "similarly to what happens in other faults"? or something similar.**

*We will change the text to include "as observed similarly in other faults" to highlight that the references refer to the general role of pressure-solution in the formation of fault fabrics.*

**line 34: it might be worth to metionting that the association of serpentinites /ultramafics with tremolite (and chlorite and talc) is well documented in the literature (e.g. Cronshaw 1923; Nishiyama 1990; Boschi et al., 2006; Bach and Klein 2009 among many others). In particular at the contact with different lithologies (rodingites but also metasediments).**

*We will revise this sentence to highlight the well-documented association of serpentinites and metasomatic products such as tremolite, talc and chlorite, particularly at the contacts between serpentinite and silicic and calcic lithologies.*

**Page 13, line 24. This statement about overpressure induced by metasomatic reactions is a bit vague. I would suggest to either remove it or present the evidence for such a phenomena (I don't think Fig. 6a is enough).**

*We will revise the statement to remove mention of fluid overpressure and instead present the observations that metasomatic reactions zones are associated with vein networks and the reaction hardening and embrittlement of metasomatised portions of the shear zone. We note that metasomatism in the Livingstone Fault forms the focus of another paper currently in revision, and we would prefer to leave details of the metasomatic reactions for that other paper.*

---

## Editor Comment (EC2) · Cristiano Collettini (Editor) · 8 Jun 2019

Dear Authors,

I have noted that in your review you have addressed al the minor points raised by the two referees. Therefore I am happy to accept your manuscript in Solid Earth.

Sincerely Cristiano Collettini

---

## Author Response (AR1)

Matthew S. Tarling
Department of Geology
University of Otago
PO Box 56
Dunedin 9054
New Zealand
tarlingmatthew@gmail.com

Dear Dr Collettini,

Thank you for the opportunity to address the weaknesses in our original manuscript that were highlighted by the reviewers. All requested changes and corrections have now been made and are incorporated in the attached manuscript. Based on the review comments, we have made revisions throughout the text and to the figures. The main amendments made to the manuscript are:

1.  We have expanded the introduction to include several other specific examples of previously-reported serpentinite-bearing shear zones
2.  We have applied a more conservative stance with regards to the temperature range estimates during deformation.
3.  We have adjusted the colour palette in Figure 1 so that there is a greater contrast between adjacent units on the map.
4.  We have increased the font size of labels and annotations in Figures 4, 5, 7, 8, 9, 10 and 11 to increase legibility.
5.  We have corrected errors in citation ordering and formatting.

We request that the detailed geologic map (figure 6) be published at high resolution at A3 size in the .pdf version of the paper.

Yours sincerely,

Matthew Tarling, on behalf of the authors

Below we include a point-by-point response to the reviewers' comments. Reviewer comments are in **bold**, author responses are in blue. Changes to the manuscript text are also highlighted by a different colour in the revised manuscript. Added text is blue and set in sans-serif, and discarded text is red and smaller in size.

Department of Geology
PO Box 56, Dunedin 9054, New Zealand
Tel +64 3 479 7519 • Fax +64 3 479 7527 • Email geology@otago.ac.nz
www.otago.ac.nz

[Figure]

**Response to Anonymous Referee #1**

We thank reviewer 1 for their constructive comments which have allowed us to clarify and improve many aspects of the manuscript. We agree with all of the reviewers' comments and have made changes to the text and figures in response to each comment.

**Specific comments (individual scientific questions/issues)**

**Page 2 (Introduction) – I recommend the authors add a sentence or two providing specific examples of other serpentinite-bearing shear zones around the world.**

As recommended, we have expanded the introduction to include several other specific examples of previously-reported serpentinite-bearing shear zones around the world (page 2, lines 14-23). Additionally, in response to comments by Reviewer Telemaco Tesei, we have expanded our brief overview of serpentinite-bearing shear zones and broadened the reference list referring to previous studies of serpentinite shear zones, including references by Maltman, 1978; Williams, 1979; Twiss and Gefell, 1990; Alexander & Harper, 1992; Gates, 1992; Bailey et al., 2000, Hirauchi and Yamaguchi, 2007, Bellot, 2008, Melosh, 2019.

**Page 9 (sections 4.3.1), Figures 9 and 10, and throughout the text – I recommend the authors specify, to the extent possible, the type(s) of serpentine minerals present in the different serpentinites. Reading between the lines, the massive serpentinite described in section 4.3.1 is likely composed of chrysotile + lizardite (+ magnetite); antigorite and other forms of serpentine are rare.**

We agree with the reviewer that we can be more precise in our definition of the different serpentine varieties, and we have put substantial effort in to developing new techniques of Raman Spectroscopy to characterise the serpentine minerals. We have added details on the types of serpentine minerals present in section 4.3.1, in figure 9 and throughout the text.

**Page 10, line 15-16. I recommend expanding the first sentence ". . . with an estimated ambient temperature during shearing of 300-350 ◦C" to articulate the constraints on the estimated temperatures, citing appropriate references. The assemblage lizardite + chrysotile can occur over a broader temperature range, but perhaps the general absence of brucite and antigorite is being used to narrow the temperature estimate.**

In response to this comment as well as a similar remark made by reviewer Telemaco Tesei, we have applied a more conservative stance with regards to the temperature range and broaden our initial estimates (page 11, lines 7-22). We have expanded our treatment of the evidence for the temperature estimates, including the instability of antigorite, the general lack of the assemblage antigorite + brucite, the metamorphic facies in the wall rocks and the dominance of a chrysotile + lizardite assemblage.

**Page 13, line 24 – What is the evidence that metasomatic reactions "can generate insitu fluid overpressures?" Would this not depend on the specific metasomatic reaction (e.g., cation exchange vs. dehydration reaction)?**

Department of Geology
PO Box 56, Dunedin 9054, New Zealand
Tel + 64 3 479 7519 • Fax + 64 3 479 7527 • Email geology@otago.ac.nz
www.otago.ac.nz

[Figure]

Yes, the reviewer is correct, and this is the subject of another paper focused on the metasomatic reactions. In the revised manuscript, we have revised this statement to remove any mention of fluid overpressure.

**Technical corrections (typos, etc.)**

**Page 3, line 5 and 14 – The rocks between the Western and Eastern Provinces are referred to as the Median "Batholith" in the text and the Median "Tectonic Zone" in Figure 1. I recommend using one or the other for consistency.**

We have revised Figure 1 such that the geological unit is consistently referred to as the Median Batholith throughout the paper.

**Page 2, line 23; page 9, line 20; page 11, line 16 - The question marks in cited references should be deleted and appropriate references added where necessary.**

All errors in referencing have been corrected.

**Page 21, Figure 1 – Several of the colours chosen for the geologic map are very similar making it difficult to distinguish some of the units (Dun Mountain terrane versus Median Tectonic Zone) particularly when similar coloured units are juxtaposed (e.g., contact between Dun Mountain terrane and Brook Street terrane). I recommend selecting different colours to ensure the geologic map can be easily interpreted.**

Following this recommendation, we have made adjustments to the colour palette of Figure 1 so that there is a greater contrast between adjacent units on the map.

**Page 32, Figure 12 – "(Chapter 4)" should be removed from box (iv) Page 27, Figure 7 –**

The erroneous label in Figure 12 box (iv) has been removed.

**This detailed geologic map should be published at the largest size possible (i.e., full page width). At its current size most of the details are not legible.**

We note that the geological map that was embedded in the submitted manuscript .pdf was very low-resolution, but that the supplementary file we submitted provided a high-resolution version. Our intention was that the detailed geologic map should be published at high resolution at A3 size in the .pdf version of the paper, and we will request this option with the journal editors and typesetters. In the online version of the paper, the map will be high resolution and zoomable, which will allow for all details to be clearly legible.

Department of Geology
PO Box 56, Dunedin 9054, New Zealand
Tel + 64 3 479 7519 • Fax + 64 3 479 7527 • Email geology@otago.ac.nz
www.otago.ac.nz

[Figure]

**Response to Telemaco Tesei**

We thank the reviewer for his detailed and constructive comments. We agree with all comments and have incorporated all the reviewers' suggestions into the revised manuscript.

**General comments: I reckon that references should be in chronological. Labels within field figures are all very small and difficult to read.**

The references have been revised and changed to be ordered chronologically in the text. We have increased the font size of labels and annotations in figures 4, 5, 7, 8, 9, 10 and 11. Unfortunately, some of the figures that were submitted in the original .pdf were converted to a lower resolution than we would have liked, which may have rendered some of the text difficult to read. We will ensure that final figure versions are submitted at highest resolution.

**Specific comments:**

**Page 2, line 16 and following. There are several more studies that document with some detail the processes and the structure of serpentine-bearing fault zones. Some potential additions to the list: Maltman, 1978; Williams, 1979; Twiss and Gefell, 1990; Alexander & Harper, 1992; Gates, 1992; Bailey et al., 2000, Hirauchi and Yamaguchi, 2007, Bellot, 2008, Melosh, 2019. In the following lines: it might be worth mentioning with some more details what these studies say about the structure and deformation of serpentinite-bearing faults. Limiting the discussion to the characteristic scaly fabric is a bit over simplistic. In the discussion, it might be useful to highlight the differences with the previous knowledge about serpentinite faults.**

We have revised the introduction to expand the overview of serpentinite-bearing shear zones worldwide and include the suggested references (page 2, lines 14-23). Additionally, in response to comments by Reviewer #1, we have provided specific reference to several examples of other serpentinite-bearing shear zones around the world.

**Page 8, line 30: there is a question mark after "Vannucchi et al., 2003" what does it means?**

Errors in referencing such as this question mark (and others throughout the text) have been corrected. The reference in question missing here is "Vannucchi 2019".

**Page 9, line 19. See also Melosh, 2019, G3**

References have been updated to include Melosh 2019.

**Page 10, line 4: How did you identify these minerals? how large are the grains?**

The minerals were identified with scanning electron microscopy energy-dispersive X-ray spectroscopy (SEM-EDS) and Raman Spectroscopy (where crystallographic symmetry permits; details in Tarling et al 2019;

Department of Geology
PO Box 56, Dunedin 9054, New Zealand
Tel +64 3 479 7519 • Fax +64 3 479 7527 • Email geology@otago.ac.nz
www.otago.ac.nz

[Figure]

Rooney et al. 2018).  Grains are typically tens of microns in size, but can be up to mm-scale. The text has been updated to include these details (page 10, lines 25-27).

**Page 10; line 16: The temperature range in which chrysotile and lizardite are stable is much wider than 300-350_C.  The absence, or instability, of antigorite may well set the high temperature boundary, but not the lower boundary. Since two close terranes have zeolite facies (T<200_C) and Prehnite-Pumpellyte facies (T<300_C) metamorphic imprint, it is possible that the Livingstone fault was active at temperatures lower than 300_C.**

Following the advice of the reviewer as well as a similar comment made by Reviewer #1, we have applied a more conservative stance with regards to the temperature range and broadened our initial estimates (11, lines 7-23. We have expanded our treatment of the evidence for the temperature estimates, including the instability of antigorite, the general lack of the assemblage antigorite + brucite, the metamorphic facies in the wall rocks and the dominance of a chrysotile + lizardite assemblage.  Additionally, the general lack of incohesive brittle fault rocks in the serpentinite shear zone would argue against any significant very shallow, low temperature deformation. Overall, the lines of evidence are compatible with a temperature range of 250-350 °C.

**Page 11, line 5: I think a paper in revision does not qualifies as previous literature.**

We have removed the paper in revision from the reference list.

**Line 8-10:  this is a very interesting observation. Do you have an estimation of the thickness of the mantle section in the other outcrops of the ophiolite adjacent to the Livingstone fault? I would be interesting to understand this change in thickness. For instance, are there any changes in kinematics or amount of displacement that could account for this change in thickness of the deformed mantle portion?**

These are interesting questions raised by the reviewer. We have not mapped in detail the ultramafic sections of the ophiolite belt adjacent to the Livingstone Fault, although crude estimates of thickness can be obtained from satellite/aerial imagery and other field observations (including other published articles and maps). Based on this, we have not noted any specific changes in shear zone kinematics that could be related to the thickness of the ultramafic portions of the ophiolite or the width of the serpentinite shear zone. However, we believe that this kind of correlation would require a more targeted project, focusing specifically on whether there is a correlation between shear zone structure/kinematics and the structure (e.g. thickness, degree of serpentinization) of the wall rocks. These questions may form the basis for future field work. Unfortunately, we lack constraints on shear zone displacement due to the absence of clear offset markers or boundaries, and thus any correlations between displacement and e.g. kinematics/wall rock structure, are currently not possible to constrain.

**Page 12, line 12: ": : :P-T conditions". Maybe a reference or two are necessary here.**

We have added references to support the notion that the interpretations reached regarding the structure, importance of pressure-solution, and metasomatic reactions are relevant to a wide range of P-T conditions.

Department of Geology
PO Box 56, Dunedin 9054, New Zealand
Tel + 64 3 479 7519 • Fax + 64 3 479 7527 • Email geology@otago.ac.nz
www.otago.ac.nz

[Figure]

**line 21. from the sentence it looks like that the references talk about pressure solution producing the scaly fabric of serpentinites. They are only some examples of work about pressure-solution weakening of faults in general. Maybe add "similarly to what happens in other faults"? or something similar.**

We have revised the text to "as well as other large-displacement fault zones with phyllosilicate-rich fault cores" to highlight that the references refer to the general role of pressure-solution in the formation of fault fabrics in phyllosilicate-rich rocks (page 13, lines 18-19).

**line 34: it might be worth to mentioning that the association of serpentinites /ultramafics with tremolite (and chlorite and talc) is well documented in the literature (e.g. Cronshaw 1923; Nishiyama 1990; Boschi et al., 2006; Bach and Klein 2009 among many others). In particular at the contact with different lithologies (rodingites but also metasediments).**

We have revised this sentence to highlight the well-documented association of serpentinites and metasomatic products such as tremolite, talc and chlorite, particularly at the contacts between serpentinite and silicic and calcic lithologies.

**Page 13, line 24. This statement about overpressure induced by metasomatic reactions is a bit vague. I would suggest to either remove it or present the evidence for such a phenomena (I don't think Fig. 6a is enough).**

We have revised the statement to remove mention of fluid overpressure and instead present the observations that metasomatic reactions zones are associated with vein networks and the reaction hardening and embrittlement of metasomatised portions of the shear zone. We note that metasomatism in the Livingstone Fault forms the focus of another paper currently in revision, and we would prefer to leave details of the metasomatic reactions for that other paper.

Department of Geology
PO Box 56, Dunedin 9054, New Zealand
Tel + 64 3 479 7519 • Fax + 64 3 479 7527 • Email geology@otago.ac.nz
www.otago.ac.nz